# HiDe: Rethinking The Zoom-IN method in High Resolution MLLMs via Hierarchical Decoupling

Xianjie Liu [* † 1]   Yiman Hu [* ‡ 1]   Yixiong Zou [2]   Liang Wu [1]   Jian Xu [1]   Bo Zheng [1]

## Abstract

Multimodal Large Language Models have made substantial progress on visual understanding tasks, yet they still perform poorly on high-resolution images. Prior work often attributes this limitation to perceptual constraints, arguing that MLLMs fail to recognize small objects and therefore rely on "zoom-in" strategies to recover fine details. In contrast, our analysis shows that the dominant failure mode is background interference rather than object size. We study the "zoom-in" operation through a **hierarchical decoupling analysis** and propose the **Hierarchical Decoupling Framework**, a training-free method that turns implicit attention into explicit region selection. HiDe first performs Token-wise Attention Decoupling to disentangle question semantics and identify the most informative tokens, then uses their attention patterns to pinpoint the corresponding visual regions. It subsequently applies Layout-Preserving Decoupling to extract these regions from cluttered backgrounds and construct a compact representation that retains key spatial structure while filtering out irrelevant context. HiDe achieves state-of-the-art results on high-resolution benchmarks like V*Bench. It boosts Qwen2.5-VL 7B and InternVL3 8B to state of the art performance, reaching 92.1% and 91.6% on V*Bench, and even surpasses reinforcement learning based methods. After optimization, HiDe reduces memory usage by 75% compared with the previous training-free approach. Code will be available at https://tennine2077.github.io/HiDe.github.io/.

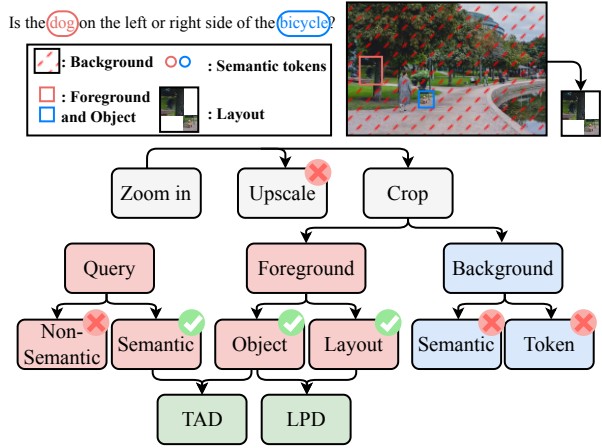

*Figure 1.* Hierarchical decoupling analysis of MLLM performance on high-resolution images. The gray, blue, red, and green parts are detailed in Sections 3.1, 3.2, 3.3, and 4, respectively.

## 1. Introduction

In recent years, Multimodal Large Language Models (MLLMs) have demonstrated remarkable capabilities in understanding and reasoning across diverse tasks such as image captioning, visual question answering, and image-text retrieval, emerging as a key driving force behind the advancement of multimodal artificial intelligence (Bai et al., 2023; Liu et al., 2023; Bai et al., 2025b; Zhu et al., 2025). With continuous improvements in model architectures and the scaling up of training data, MLLMs have achieved significant progress in handling complex semantics and achieving fine-grained cross-modal alignment (Feng et al., 2026b). However, recent studies (Zheng et al., 2026; Zhang et al., 2025a; Shen et al., 2025; Wu & Xie, 2024; Wang et al., 2025; Li et al., 2025) have revealed that existing approaches still fall short of expectations in handling high-resolution image tasks, as evidenced by their suboptimal performance on benchmarks such as V*Bench (Wu & Xie, 2024).

To enhance the model's perception of image details, existing methods often adopt a "locate-then-zoom-in" strategy. Training-based approaches, such as Supervised Fine-Tuning (SFT) (Shao et al., 2024) or Reinforcement Learning (RL) (Zheng et al., 2026; Zhao et al., 2025), can guide models to

---
[*]Equal contribution , [†]Work done during an internship at Alibaba., [‡]Project leader. [1]Alimama Tech, Taobao & Tmail Group of Alibaba [2]Huazhong University of Science and Technology. Correspondence to: Yiman Hu <huyiman.hym@alibaba-inc.com>, Liang wu <wuliang.wu@taobao.com.>, Jian Xu <xiyu.xj@taobao.com.>, Bo Zheng <bozheng@alibaba-inc.com>.

*Proceedings of the 43rd International Conference on Machine Learning*, Seoul, South Korea. PMLR 306, 2026. Copyright 2026 by the author(s).

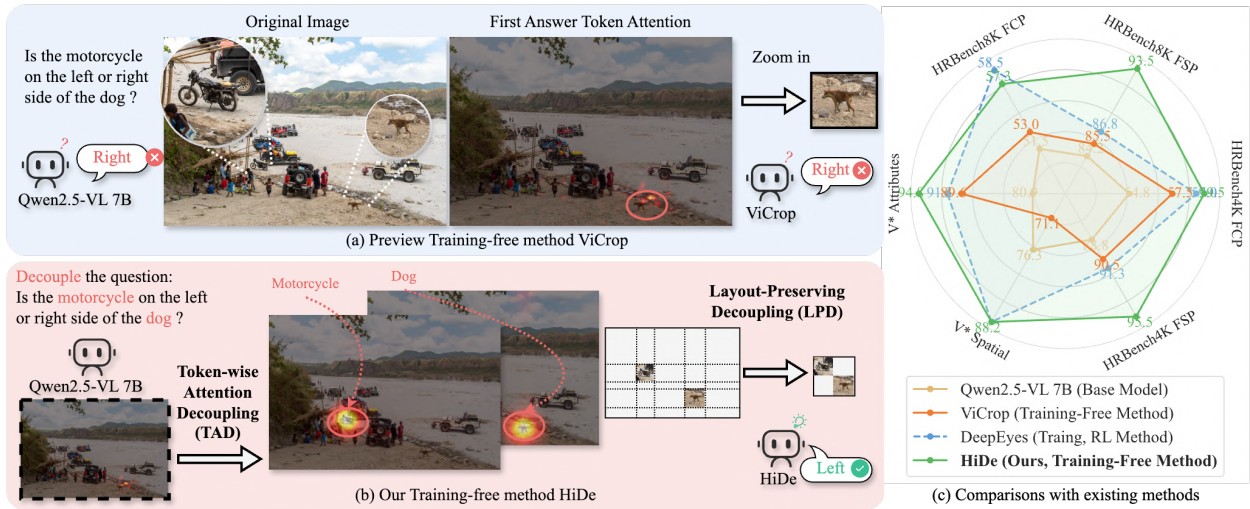

*Figure 2.* (a) Previous methods struggle to locate objects. (b) HiDe precisely locates objects and keeps relative positions. (c) HiDe outperforms previous training-free and beats the trained one.

identify relevant regions. However, they suffer from critical drawbacks (Zhai et al., 2023; Yue et al., 2026), including high costs, lengthy training, and a lack of cross-architecture transferability, which severely limits their scalability and practical deployment. Training-free methods (Zhang et al., 2025a; Shen et al., 2025; Li et al., 2025), which automatically locate regions by analyzing attention or performing tree-based search, are often inefficient at inference due to multiple forward passes and exhibit high miss rates, especially when handling multiple objects, limiting practicality.

However, we find that the widely used "zoom-in" strategy conflates two operations, upscaling and cropping, and that upscaling alone does not improve MLLM performance. To trace the true bottlenecks and isolate what actually improves performance, we conduct a **hierarchical decoupling analysis** (Fig. 1) starting from the zoom-in operation:

(i) Zoom-in → upscale and crop. Quantitative results show that scaling the full image provides little benefit, while cropping accounts for most of the improvement.

(ii) Crop → foreground and background. We split the cropped area into foreground and background, and further decompose the background into **semantic distractors** and **token-level redundancy**. Both factors introduce substantial interference and degrade MLLM reasoning.

(iii) Question text → semantic and non-semantic tokens. By separating semantic from non-semantic query tokens, we find that semantic tokens primarily drive vision language alignment and enable accurate localization of the relevant regions.

(iv) Foreground → object appearance and spatial layout. Using objects indicated by semantic tokens, we show that modeling their relative spatial layout is critical for accurate

judgments beyond recognizing object appearance.

This structured decomposition reduces the problem to its minimal factors and allows us to selectively leverage the effective components while discarding the detrimental ones, thereby improving MLLMs' performance on high-resolution understanding.

Guided by these insights, we propose **HiDe**, a Hierarchical Decoupling method for precise and structured visual representation. As shown in Fig. 2, HiDe consists of two modules. **Token-wise Attention Decoupling (TAD)** isolates fine-grained token region alignments by denoising token-level attention maps, mitigating background noise such as visual sinks, and enabling accurate localization. **Layout-Preserving Decoupling (LPD)** then binarizes the purified maps to separate relevant regions from clutter and reconstructs a compact representation via grid-based aggregation that preserves the key spatial layout.

We conduct experiments on high-resolution datasets V* (Wu & Xie, 2024), HRBench-4K, and HRBench-8K (Wang et al., 2025), with prominent MLLMs like Qwen2.5-VL, Qwen3-VL and InternVL3. HiDe surpasses existing training-free methods and even outperforms RL-trained approaches, achieving state-of-the-art results on both single-object and multi-object benchmarks. In addition, through a simple yet effective engineering modification, we reduce the peak memory usage from 96 GB to 20 GB (75%), improving the practical applicability. Our contributions are summarized as follows:

- We perform a hierarchical decoupling of existing zoom-in approaches, revealing background distraction as the root cause of MLLMs' limitations on high-resolution images and isolating the components that genuinely

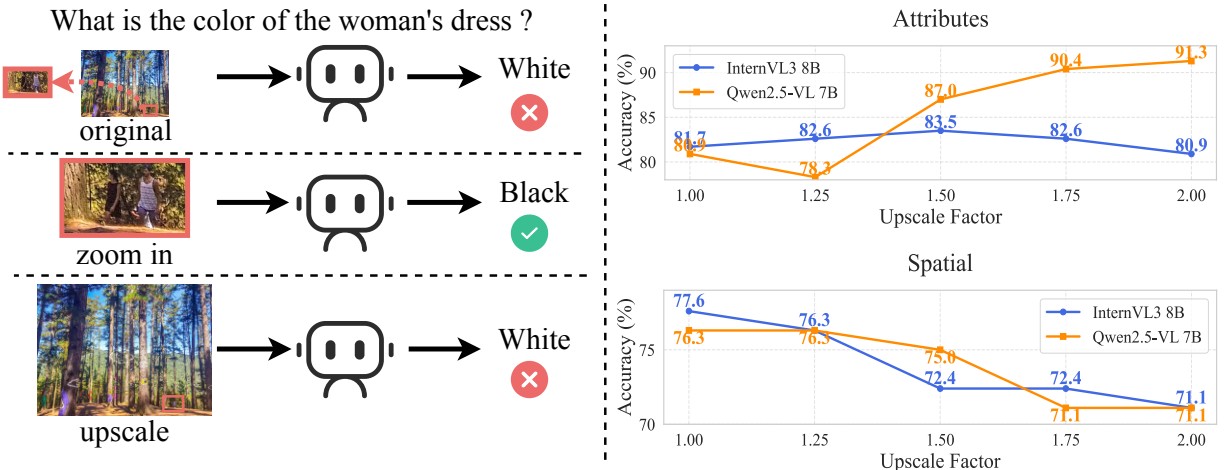

*Figure 3.* Left: A contradictory example comparing the inference results of zoom-in and simple resolution upscaling at the same upscale factor. Right: Performance curves showing the impact of resolution scaling on two models across two tasks—Attributes for single-object tasks and Spatial for multi-object tasks.

drive performance gains.

- We propose HiDe, a hierarchical decoupling framework (TAD + LPD) that extracts precise, compact, and structurally coherent visual representation.

- Our method achieve state-of-the-art accuracy on both single- and multi-object tasks with only a small increase in inference overhead, and reduce memory footprint by 75% compared to previous training-free methods, enhancing the practicality of our method.

## 2. Related Works

**Multimodal Large Language Models (MLLMs).** Multimodal Large Language Models (MLLMs). MLLMs are foundational for diverse vision-language tasks. Early fixed-resolution models (Liu et al., 2023; Bai et al., 2023; Li et al., 2023a; 2022; Liu et al., 2024b;a) often sacrifice fine-grained details during resizing. To address this, dynamic-resolution MLLMs process images at original scales to preserve spatial fidelity via adaptive visual tokens. For instance, InternVL3 (Zhu et al., 2025; Chen et al., 2024a; Wang et al., 2024b; Gao et al., 2024; Chen et al., 2024b;c) utilizes patch-based encoding with standard ViT (Dosovitskiy et al., 2021), while Qwen-VL (Bai et al., 2025a;b; Wang et al., 2024a; Bai et al., 2023) employs an end-to-end native-resolution ViT. Our work advances the understanding of visual information utilization and proposes a scalable, training-free method to enhance perceptual capabilities, providing an orthogonal and effective complement to existing approaches.

**High-Resolution Visual Question Answering (HR-VQA).** HR-VQA (Wu & Xie, 2024; Wang et al., 2025; Zhang et al., 2025b) shifts the focus of standard VQA (Li et al., 2023b; Kazemzadeh et al., 2014) from holistic scenes to fine-grained details, where current MLLMs often underperform. Existing solutions follow two main paths: (1) training-based methods using SFT (Shao et al., 2024; Wu & Xie, 2024) or RL (Zheng et al., 2026; Zhao et al., 2025; Chen et al., 2025), which risk catastrophic forgetting (Zhai et al., 2023) and reduced robustness (Yue et al., 2026); and (2) training-free strategies using attention-based cropping (Zhang et al., 2025a) or tree search (Li et al., 2025; Wang et al., 2025), which are often inefficient, incompatible with FlashAttention (Dao et al., 2022), or prone to missing targets (Feng et al., 2026a). Our work introduces a novel training-free mechanism that overcomes these limitations via efficient multi-target localization, offering a flexible and powerful framework for complex visual tasks.

## 3. Hierarchical Decoupling Analysis

In this section, we conduct a hierarchical decoupling analysis to identify the root causes of MLLMs' underperformance on high-resolution images and outline key insights for improving their effectiveness. The decoupling framework is illustrated in Fig. 3. All experiments in this section are conducted on the V* dataset(Wu & Xie, 2024).

### 3.1. Decoupling the Zoom-in Operation

Previous works suggest that small objects can impair model's judgment and propose a zoom-in method (crop + upscale) to enhance perception. To isolate the effects of cropping and magnification, we uniformly upscale each image without cropping. From the left panel of Fig. 3, we observe that even when the target object size matches the size achieved by zoom-in, the model still fails.

To achieve a comprehensive analysis, we upscale dataset

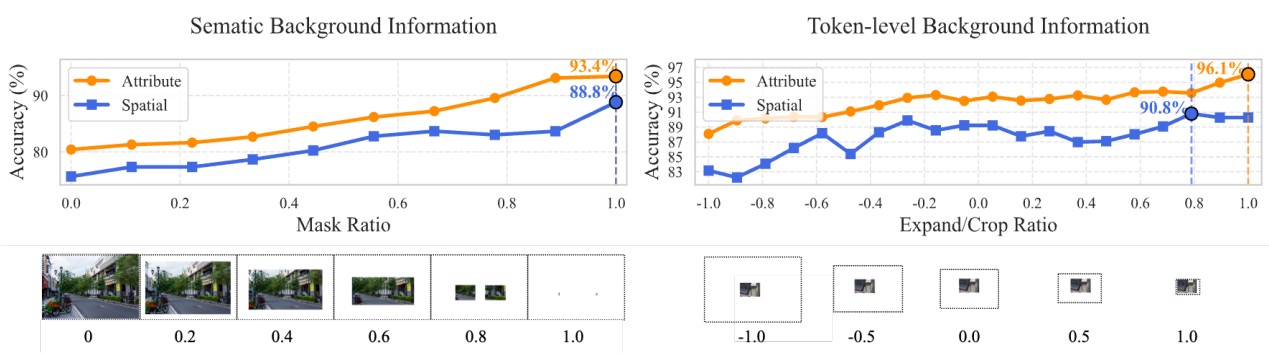

*Figure 4.* **Background Information Ablation Experiments.** Left: Model accuracy increases as the mask ratio of background semantic information rises. Right: Model accuracy improves as the number of background tokens decreases. Each point represents the average accuracy over 10 steps.

images by various factors and evaluate the performance of Qwen2.5-VL and InternVL3. Results in the right panel of Fig. 3 demonstrate that **simply enlarging the object does not deliver stable gains**; on multi-object tasks, magnification can even hurt performance. This indicates that **zoom-in works primarily because cropping removes large amounts of irrelevant high-resolution background, not because the upscaling makes the model "sees more clearly"**. This hypothesis aligns with findings in text-only settings(Liu et al., 2025): critical evidence can be obscured by redundant information in long texts.

### 3.2. Decoupling the crop operation

We further decouple the effects of cropping into two components: removing background semantics and reducing token-level redundancy to evaluate their individual contributions.

**(1) Removing background semantics.** Using transparent masks at fixed resolution, we progressively mask non–ground-truth (non-GT) regions of each image. The mask ratio in [0, 1] denotes the fraction of background removed (1: all non-GT masked; 0: original). With 100 granularity steps, we evaluate Qwen2.5-VL at each step. As shown in Fig. 4 left, performance increases monotonically with mask ratio on both single and multi-object tasks. This demonstrates that complex background semantics significantly distract MLLMs.

**(2) Reducing token-level redundancy.** Building on experiments in (1), masked images still include large transparent areas that the vision encoder turns into redundant tokens. To investigate the effects of them, we trim these areas and, for contrast, also expand them. The results in Fig. 4 right shows that, as token-level redundancy decreases, accuracy improves; expanding transparent padding produces the opposite trend. This indicates that redundant tokens not only increase computational overhead but also interfere with MLLM attention and reasoning.

Together, these results show that **MLLMs are sensitive to both semantic background distractors and excess background tokens**. Cropping helps because it simultaneously removes semantic clutter and cuts token load.

### 3.3. Decoupling Key objects

Having established that background information is the core bottleneck for MLLMs in high-resolution image understanding, we now focus on accurately isolating foreground content. We decouple the task into two components: object localization and modeling of their relative spatial layout.

**(1) Leveraging Token-level Attention for Accurate Region Proposals.** Conventional methods localize regions using attention from the first answer token, assuming it captures all key semantics. However, this approach faces two issues (Fig. 5a): (a) it often misses objects in multi-target scenes, and (b) it incorporates non-semantic noise. We observe that visual cues vary by token type (Fig. 5b): while function words disperse attention, semantic nouns focus on compact, informative regions. Leveraging these token-level differences enables more precise and complete region identification. The analysis of Internvl3 in Appendix Sec. C.

Therefore, we decouple the attention of the first answer token by categorizing query text tokens into semantic and non-semantic groups. We then analyze their respective focus on image ground-truth regions by examining the cross-attention maps.

Specifically, for each text token $t_i$, we first compute its attention over the entire image, represented as a sequence of all image patch tokens $\mathcal{P} = \{p_1, p_2, \ldots, p_K\}$, where $K$ is the total number of patches. The attention weight distribution of $t_i$ across the full image at layer $l$ is computed using the standard scaled dot-product attention:

$$A_i^{(l)} = \text{softmax}\left(\frac{t_i^{(l)} \cdot \mathcal{P}^\top}{\sqrt{d_k}}\right), \qquad (1)$$

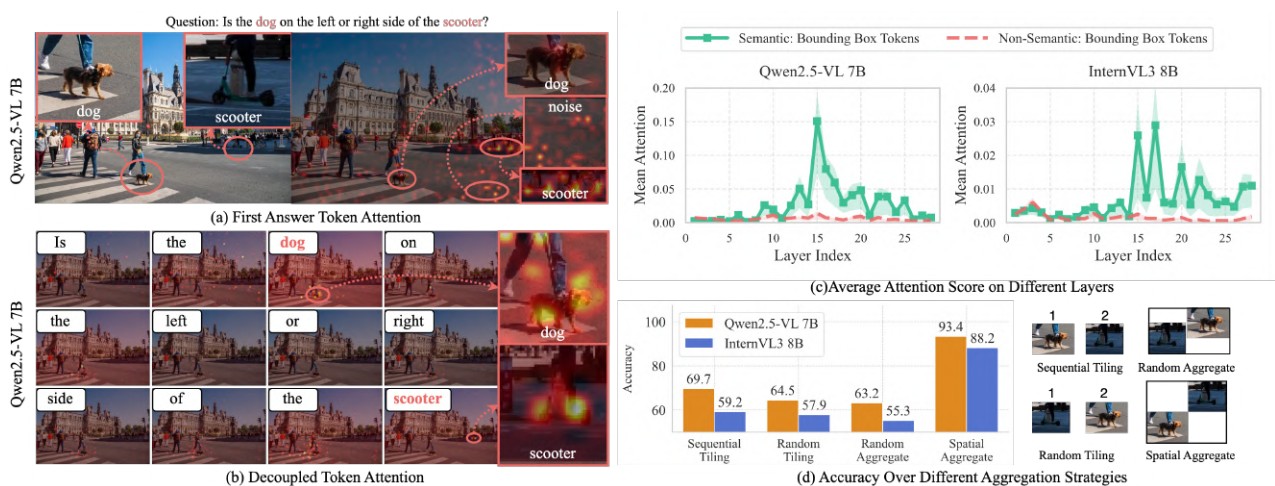

*Figure 5.* (a, b) Visualization of attention maps. (a): Attention map from the first generated answer token, miss some target regions and has noise. (b): Attention maps for every input question token, accurately localizing target regions based on corresponding tokens. (c): Relative attention to the bounding box areas across the layers for Qwen2.5-VL and InternVL3. (d): Accuracies of different aggregate methods, Spatial Aggregate is the best strategy.

where the $t_i^{(l)}$ is the hidden states at layer $l$, and $d_k$ is the attention head dimension.

Given the ground-truth region $R$, which corresponds to a subset of spatially contiguous patches $\{p_j \mid j \in \mathrm{Idx}(R)\} \subset \mathcal{P}$, we extract the attention scores over these patches:

$$A(t_i, R)^{(l)} = \left\{ A(t_i, p_j)^{(l)} \mid p_j \in R \right\}, \qquad (2)$$

where the $p_j^{(l)}$ is the hidden states at layer $l$. The average attention score of semantic text tokens $T_{\mathrm{sem}} = \{t_1, t_2, \ldots, t_M\}$ on the GT region is then defined as:

$$\bar{A}_{T_{\mathrm{sem}}, R}^{(l)} = \frac{1}{|T_{\mathrm{sem}}| \cdot |R|} \sum_{t_i \in T_{\mathrm{sem}}} \sum_{p_j \in R} A(t_i, p_j)^{(l)}, \quad (3)$$

where $|T_{\mathrm{sem}}|$ is the number of semantic tokens and $|R|$ is the number of image patch tokens within the GT region. This metric quantifies how well semantic words attend to relevant visual content. Non-semantic token sequences are evaluated analogously.

Average $\bar{A}$ scores across layers (Fig. 5c) show that semantic tokens attend significantly more to GT regions than non-semantic ones, confirming that **token-level attention decoupling yields more accurate region proposals**. Mid-layers provide the strongest signal, likely because deeper layers shift focus from fine-grained visual details toward semantic integration and reasoning.

**(2) Modeling Spatial Layouts for Regions Representation.** After acquiring multiple target regions through decoupled attention, how should they be fed back to the MLLM? We evaluate four strategies on the Spatial subset of V*: Sequence Tiling: tiling GT regions in scan order, top-left to bottom-right. Random Tiling: same tiles, shuffled order. Spatial Aggregate: recomposing GT regions while preserving relative positions. Random Aggregate: recomposing then shuffling positions. Across both Qwen2.5-VL and Intern3VL, preserving relative spatial layout (Spatial Aggregate) markedly improves performance (Fig. 5 (d)), **underscoring the importance of spatial layout modeling when aggregating target region.** We detail the layout-preserving aggregation in Sec. 4.2.

## 4. HiDe: Hierarchical Decoupling for Precise and Compact

Based on the analysis in Sec. 3, we propose the **Hierarchical Decoupling Framework (HiDe)** to enhance MLLMs for high-resolution, fine-grained tasks (Fig. 6). HiDe comprises two key components: First, **Token-wise Attention Decoupling (TAD)** maps semantic information to visual regions via internal attention. It purifies attention maps by subtracting a ques noise, effectively isolating discriminative signals from background activations. Second, **Layout-Preserving Decoupling (LPD)** converts these purified signals into bounding boxes and decouples localized regions from the background. These regions are reconstructed into a compact image that eliminates irrelevant information while strictly preserving the targets' relative spatial configuration.

### 4.1. Token-wise Attention Decoupling (TAD)

As established in Sec. 3, individual semantic tokens can provide precise visual cues that are often diluted in the model's aggregate attention. To exploit this, our framework's first stage performs signal-level decoupling. This process, which

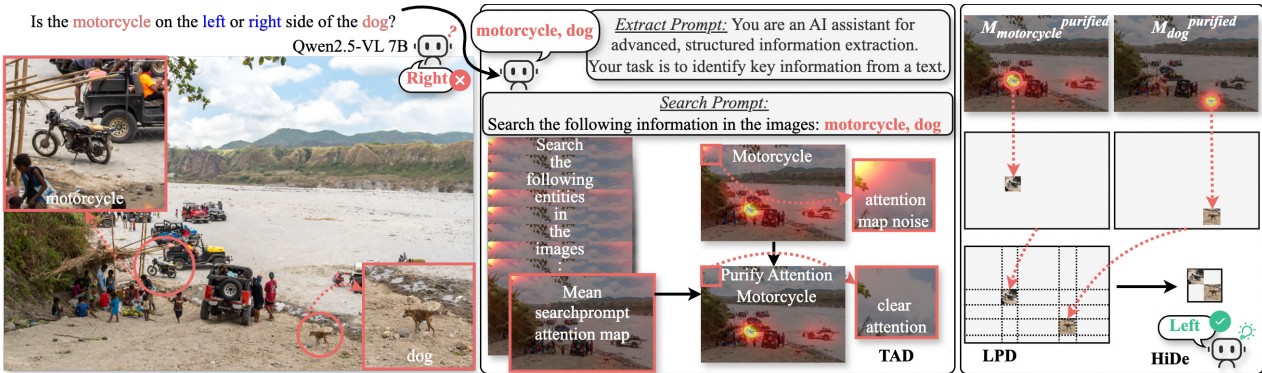

Figure 6. **The framework of HiDe.** Pure attention maps are obtained using TAD, followed by LPD to generate compact target region image. Both the target region image and the original image are fed into the MLLM to get the correct answer.

we term **Token-wise Attention Decoupling (TAD)**, is designed to isolate these fine-grained alignments and purify them from background noise for accurate localization.

We first extract key tokens $\{t_i\}$ (Fig. 6) and compute their $H \times W$ raw attention maps $A_i$.

To combat "visual sink" noise (Kang et al., 2025), we smooth $A_i$ via $\tilde{A}_i = G_\sigma * A_i$ and subtract a background noise prior estimated from irrelevant tokens in a generic "search" prompt:

$$M_i^{\text{purified}} = \mathcal{N}(\tilde{A}_i) - \mathbb{E}_{q \in \text{SearchPrompt}}\left[\mathcal{N}(\tilde{A}_q)\right], \quad (4)$$

where $\mathcal{N}(x) = (x - \min x)/(\max x - \min x)$ denotes min-max normalization. This process removes non-discriminative patterns, highlighting regions uniquely relevant to each token.

Naively computing attention weights incurs substantial GPU memory overhead. To address this, we propose an efficient scheme integrated with FlashAttention (Dao et al., 2022): we first perform a forward pass using FlashAttention, then compute attention weights precisely only for the $n$ text queries with CPU offloading. This strategy reduces peak GPU memory usage from over 96GB to 20GB.

### 4.2. Layout-Preserving Decoupling (LPD)

After obtaining a set of purified attention maps $\{M_i^{\text{purified}}\}$, the next step is to extract the key regions from the attention signals. This entire stage is orchestrated by our **Layout-Preserving Decoupling (LPD)** mechanism, which transforms these abstract attention signals into a concrete, compact image representation. LPD operates in two stages: (i) discretizing attention signals into spatial regions, and (ii) recomposing these regions on a new canvas while preserving relative spatial configurations.

**Region Discretization.** Each map $M_i^{\text{purified}}$ is binarized via

a threshold $\alpha \in [0, 1]$ after min-max normalization:

$$M_i = \mathbb{I}\left(\mathcal{N}(M_i^{\text{purified}}) > \alpha\right), \quad (5)$$

where $\mathbb{I}(\cdot)$ is the indicator function. LPD then extracts connected components $\{c_j\}$ to form bounding boxes $b_j = (\min x_p, \min y_p, \max x_p, \max y_p)$ for $p \in c_j$. The resulting set $\mathcal{B} = \bigcup_i \{b_j\}$ grounds all relevant semantic concepts.

**Spatial Reconstruction.** To avoid destroying vital spatial relationships or retaining background interference (Sec. 3.3), LPD employs a grid-based reconstruction (Alg. 13). We define a grid from the sorted coordinates of $\mathcal{B}$, yielding lines $S_X$ and $S_Y$. Indicators $I_c(i)$ and $I_r(j)$ identify content-bearing columns and rows. A pixel $(x, y)$ is then mapped to $(x', y')$ in the compact image $I_{\text{compact}}$ via:

$$(x', y') = \left((x - s_{x,i}) + \sum_{l=0}^{i-1} I_c(l) \cdot \Delta x_l, \right.$$
$$\left. (y - s_{y,j}) + \sum_{l=0}^{j-1} I_r(l) \cdot \Delta y_l\right), \quad (6)$$

where $\Delta x_l = s_{x,l+1} - s_{x,l}$ and $\Delta y_l = s_{y,l+1} - s_{y,l}$. This transformation effectively stitches content cells while discarding empty regions. Finally, the original image, $I_{\text{compact}}$, and the question are fed to the model for the final answer. The python-style pipeline is in Appendix Sec. J.

## 5. Experiments

### 5.1. Experimental Settings

**Implementation details.** We used a computing card PPU-ZW810E with maximum memory capacity of 96 GB. We fix the temperature of model outputs to 0 to ensure deterministic responses given the same input, thereby avoiding inconsistent results caused by randomness. For InternVL3 (Zhu et al., 2025), we set the number of image input token blocks

*Table 1.* We report answer accuracy for MLLMs on HR-VQA tasks. The best score is highlighted in **bold**, and the second is underlined.

| Method | Model | Train Free | V* | | | HRBench4K | | | HRBench8K | | |
|---|---|---|---|---|---|---|---|---|---|---|---|
| | | | Attr | Spatial | Avg | FSP | FCP | Avg | FSP | FCP | Avg |
| - | GPT-4o | - | - | - | 66.0 | 70.0 | 48.0 | 59.0 | 62.0 | 49.0 | 55.5 |
| - | OpenAI o3 | - | - | - | **95.7** | - | - | - | - | - | - |
| - | InternVL3 8B | - | 81.7 | 78.9 | 80.6 | 82.8 | 58.8 | 70.8 | 80.0 | 59.8 | 69.9 |
| ViCrop | InternVL3 8B | ✓ | 88.7 | 75.0 | 83.3 | 88.0 | 57.0 | 72.5 | 82.8 | 54.8 | 68.8 |
| **HiDe** | InternVL3 8B | ✓ | 92.2 | **90.8** | 91.6 | 90.0 | 63.5 | 76.8 | 92.2 | 62.0 | 77.1 |
| Δ(*vs* InternVL3 8B) | - | | +10.5 | +12.1 | +10.0 | +6.2 | +4.7 | +6.0 | +12.2 | +2.2 | +7.2 |
| - | Qwen2.5-VL 3B | - | 80.9 | 61.8 | 73.3 | 83.0 | 50.3 | 66.6 | 79.8 | 45.0 | 62.4 |
| ViCrop | Qwen2.5-VL 3B | ✓ | 81.7 | 65.8 | 75.4 | 86.3 | 49.8 | 68.0 | 80.3 | 45.5 | 62.9 |
| **HiDe** | Qwen2.5-VL 3B | ✓ | 85.2 | 72.4 | 80.1 | 87.8 | 51.5 | 69.6 | 88.3 | 51.3 | 69.8 |
| Δ (*vs* Qwen2.5-VL 3B) | - | | +4.3 | +10.6 | +6.8 | +4.8 | +1.2 | +3.0 | +8.5 | +6.3 | +7.4 |
| - | Qwen2.5-VL 7B | - | 80.9 | 76.3 | 79.1 | 88.8 | 54.8 | 71.8 | 84.2 | 51.5 | 67.9 |
| ViCrop | Qwen2.5-VL 7B | ✓ | 89.6 | 71.1 | 82.2 | 90.5 | 57.5 | 74.0 | 85.5 | 53.0 | 69.3 |
| DeepEyes | Qwen2.5-VL 7B | ✗ | 91.3 | 88.2 | 90.1 | 91.3 | 59.0 | 75.1 | 86.8 | 58.5 | 72.6 |
| **HiDe** | Qwen2.5-VL 7B | ✓ | **94.8** | 88.2 | 92.1 | **95.5** | 59.5 | 77.5 | 93.5 | 57.3 | 75.4 |
| Δ (*vs* Qwen2.5-VL 7B) | - | | +13.9 | +11.9 | +13.0 | +6.7 | +4.7 | +5.7 | +9.3 | +5.8 | +7.5 |
| - | Qwen2.5-VL 32B | - | 87.8 | 88.1 | 87.9 | 89.8 | 58.0 | 73.9 | 84.5 | 56.3 | 70.4 |
| ViCrop | Qwen2.5-VL 32B | ✓ | 90.4 | 85.5 | 88.5 | 90.5 | 60.0 | 74.6 | 88.8 | 54.3 | 71.5 |
| **HiDe** | Qwen2.5-VL 32B | ✓ | 91.3 | 89.5 | 90.6 | 92.0 | 60.0 | 76.2 | 90.5 | 57.5 | 74.0 |
| Δ (*vs* Qwen2.5-VL 32B) | - | | +3.5 | +1.4 | +2.7 | +2.2 | +2.0 | +2.3 | +6.0 | +1.2 | +3.6 |
| - | Qwen3-VL 8B | - | 87.0 | 78.9 | 83.8 | 90.2 | 63.7 | 77.0 | 84.0 | 62.3 | 73.1 |
| ViCrop | Qwen3-VL 8B | ✓ | 89.6 | 75.0 | 83.8 | 91.8 | 64.3 | 78.0 | 88.0 | 63.3 | 75.6 |
| **HiDe** | Qwen3-VL 8B | ✓ | 92.9 | 85.5 | 89.5 | 93.5 | **69.0** | **81.2** | **96.5** | **71.2** | **83.9** |
| Δ (*vs* Qwen3-VL 8B) | - | | +5.9 | +6.6 | +5.7 | +3.3 | +5.3 | +4.2 | +12.5 | +8.9 | +10.8 |

to its maximum supported value 36. For Qwen2.5-VL (Bai et al., 2025b) and Qwen3-VL (Bai et al., 2025a), we set the maximum number of image input pixel tokens to 16384 and set the minimum number of image input pixel tokens to 256, aligning with the minimum image resolution 448×448 of InternVL3. For Qwen2.5-VL 7B and Qwen3-VL 8B, we use layer 15, set $\sigma = 3$ and $\alpha = 0.7$. For InternVL3 8B, we use layer 17, set $\sigma = 2$ and $\alpha = 0.6$.

## 5.2. Main Results

To ensure a comprehensive evaluation, we select representative systems: (1) SOTA proprietary models, including GPT-4o (OpenAI., 2024) and OpenAI o3 (OpenAI., 2025); and (2) widely adopted open-source models, Qwen2.5-VL (Bai et al., 2025b) and InternVL3 (Zhu et al., 2025). In addition, we compare our method against leading approaches in high-resolution visual understanding, including Deep-Eyes (Zheng et al., 2026), which is based on reinforcement learning, and ViCrop (Zhang et al., 2025a), a training-free method that relies on cropping and zooming. For open-source methods, except for Deepeyes whose results are directly taken from its original paper, all other methods have been re-evaluated. Table 1 summarizes the performance of all these models and methods across three challenging benchmarks: V*Bench (191 samples) (Wu & Xie, 2024),

HRBench4K (800 samples) and HRBench8K (800 samples) (Wang et al., 2025). For larger datasets such as POPE (9,000 samples) (Li et al., 2023b), MME-RealWorld-Lite (1,919 samples) and MME-RealWorld-English (23,609 samples) (Zhang et al., 2025b), we present the results based on Qwen2.5-VL 7B in the Appendix Sec. D.

The experimental results demonstrate that our method achieves the best overall performance among all competing approaches. It exhibits strong robustness and generalization across diverse model architectures and task types. It delivers consistent improvements not only in single-object recognition tasks such as attribute prediction (Attr) and fine-grained semantic parsing (FSP), but also in more complex multi-object scenarios involving spatial reasoning (Spatial) and compositional understanding (FCP). Moreover, unlike methods that require costly fine-tuning or iterative reasoning strategies such as tree search, HiDe is training-free and provides a plug-and-play capability. We shown more experiments like low-resolution and only the compact input experiments in Appendix Sec. F and Table 12. More case and failure case in Appendix Sec. H and Appendix Sec. I.

## 5.3. Visual Attention Regions Comparison

To illustrate TAD's localization accuracy, Fig. 7 compares our purified attention maps (overlaid for multi-target cases)

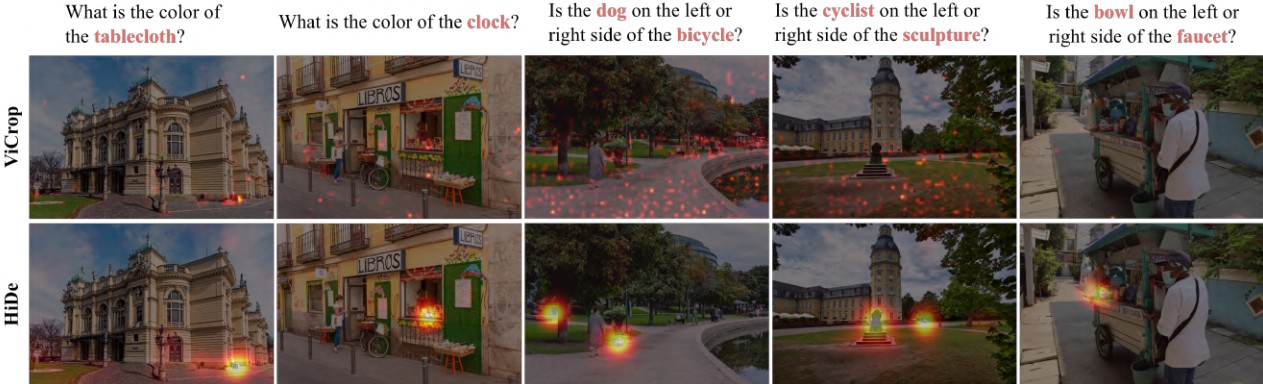

*Figure 7.* Comparative visualization of attention weights for HiDe and ViCrop.

*Table 2.* Different methods for extracting the region of interest.

| Method | Attr | Spatial | Avg |
|---|---|---|---|
| Qwen2.5-VL 7B (Base) | 80.9 | 76.3 | 79.1 |
| Base + predicted BBox | 82.6 | 82.9 | 82.7 |
| Base + Question token | 90.4 | 81.6 | 86.9 |
| Base + Decomposing semantic | 93.9 | 86.8 | 91.1 |
| Base + **TAD** | **94.8** | **88.2** | **92.1** |

*Table 3.* Different methods for connecting the region of interest.

| Method | Attr | Spatial | Avg |
|---|---|---|---|
| Qwen2.5-VL 7B (Base) | 80.9 | 76.3 | 79.1 |
| Base + Sequential | **94.8** | 73.7 | 86.4 |
| Base + Masking | 87.8 | 84.2 | 86.4 |
| Base + LPD w/o Compaction | 87.8 | **88.2** | 88.0 |
| Base + **LPD** | **94.8** | **88.2** | **92.1** |

with ViCrop's first-token attention. TAD precisely captures targets in both single- and multi-object scenarios, whereas ViCrop's maps are scattered and fail to localize all relevant instances. This underscores TAD's superior precision and semantic alignment. And we show the raw attention of Qwen2.5-VL 7B in Appendix Sec. B.

### 5.4. Ablation Study

Due to the scarcity of high-resolution VQA benchmarks, we perform our ablations on V*Bench; to avoid dataset bias, the optimal configurations identified here are fixed as defaults and applied consistently across all other datasets. We conduct ablation studies on V*Bench using Qwen2.5-VL 7B to evaluate two core components: TAD and LPD. All the ablation using original image and target regions images. First, for critical region extraction, we compare four progressive approaches as shown in Table 2: (1) Model-predicted BBox: bounding boxes predicted by a base model; (2) Question every token: attention from all question tokens without refinement; (3) Decomposing semantic units: attention based on extracted key information, corresponding to the initial step of TAD without purification; (4) TAD: our full token-level decoupling pipeline with attention purification. To ensure fair comparison, all detected regions are reconstructed using the LPD mechanism. Second, we evaluate four region concatenation strategies, starting from identical purified regions generated by our TAD. Results are summarized in Table 3: (1) Sequential Concatenation:

*Table 4.* Performance comparison on V*Bench. "OOM" denotes an Out of Memory error.

| Method | GPU Memory | Inference Time |
|---|---|---|
| Qwen2.5-VL 7B | ~20 GB | ~7 min |
| DeepEyes | ~20 GB | ~40 min |
| ViCrop | >96 GB (OOM) | N/A |
| ViCrop* | ~20 GB | ~28 min |
| **HiDe** | ~20 GB | ~14 min |

regions are cropped and concatenated in sequence; (2) Masking: background is masked as transparent while preserving original layout and resolution; (3) LPD w/o Compaction: LPD preserves relative spatial layout but pads output to the original image size; (4) LPD (Ours): our method, achieving both layout preservation and compact, interference-free representation. Ablation on hyper-parameters $\sigma$ and $\alpha$ and layers is provided in Appendix Sec. A and Sec. E. Quantitative evaluations of TAD accuracy and a study on the role of global context are provided in Appendix Sec. 5.6.

### 5.5. Efficiency Assessment

We evaluate computational efficiency on V*Bench using a single PPU-ZW810E accelerator (96 GB max memory), with results summarized in Table 4. While the baseline Qwen2.5-VL 7B with FlashAttention-v2 (Dao et al., 2022) naturally operates within ~20 GB VRAM, the original ViCrop routinely triggers OOM failures. Compared to **ViCrop**,

*Table 5.* Quantitative study of HiDe generated bounding boxes on V*Bench

| Metric | Method | Attr | Spatial | Avg | Method | Attr | Spatial | Avg |
|---|---|---|---|---|---|---|---|---|
| IoU | Internvl3 8B | 0.043 | 0.026 | 0.036 | Qwen2.5-VL 7B | 0.024 | 0.027 | 0.025 |
| Precision | ViCrop | 0.044 | 0.027 | 0.037 | ViCrop | 0.024 | 0.028 | 0.026 |
| Recall | Single box | 0.747 | 0.368 | 0.596 | Single box | 0.783 | 0.465 | 0.657 |
| IoU | Internvl3 8B | **0.092** | 0.072 | 0.084 | Qwen2.5-VL 7B | 0.085 | **0.096** | **0.089** |
| Precision | **HiDe** | **0.096** | 0.075 | 0.088 | **HiDe** | 0.087 | **0.100** | **0.093** |
| Recall | Multiple boxes | 0.816 | 0.806 | 0.812 | Multiple boxes | **0.931** | **0.889** | **0.914** |

our efficiency stems from two key algorithmic and engineering designs: (i) **Targeted tensor offloading**, which executes a single FlashAttention pass and precisely computes attention weights only for $n$ semantic query tokens, strategically offloading intermediate tensors to system RAM. This bypasses the 96 GB bottleneck, reduces peak memory to $\sim$20 GB (a 75% reduction), and alleviates GPU bandwidth pressure; (ii) **One-shot region identification**, which replaces ViCrop's costly two-pass sliding-window search with a deterministic TAD→LPD pipeline, halving inference latency from $\sim$28 min (ViCrop*) to $\sim$14 min.

Compared to RL methods like **DeepEyes** ($\sim$40 min), HiDe offers two inherent advantages. First, as a **training-free** framework, it completely eliminates costly reward modeling, gradient updates, and associated training resource waste. Second, during inference, RL methods rely on generating lengthy "Think" chains to guide step-by-step reasoning, which drastically prolongs generation latency. HiDe bypasses this overhead by directly constructing a compact, layout-preserving visual representation, enabling rapid answer generation without iterative token sampling. These combined gains allow seamless deployment on commodity hardware while maintaining SOTA accuracy.

### 5.6. Experiment of HiDe generated bounding boxes

To quantitatively assess HiDe's bounding box accuracy, we computed IoU, precision, and recall (higher is better) against ground-truth boxes on V*Bench, comparing with ViCrop. As shown in Table 5, the results strongly support our claims: our token-level attention decoupling (TAD) effectively localizes query-relevant regions, enabling more accurate background removal and performance gains.

## 6. Conclusion

In this paper, we introduce HiDe, a novel framework that rethinks the conventional zoom-in paradigm for high-resolution multimodal large language models. Through a rigorous hierarchical decoupling analysis, we identify background token redundancy as the primary constraint that dilutes model attention and obscures fine-grained visual cues.

This interference fundamentally limits multi-object reasoning and renders naive resolution scaling largely ineffective. To address this challenge, HiDe systematically disentangles visual perception into two complementary stages. The Token-wise Attention Decoupling module isolates query-specific semantic signals from noisy attention maps to enable precise target localization without relying on iterative searches. The Layout-Preserving Decoupling module subsequently extracts these critical regions and reconstructs them into a compact visual representation that strictly maintains original spatial relationships while eliminating irrelevant contextual clutter. Operating as a strictly training-free and architecture-agnostic solution, HiDe integrates seamlessly into existing models as an efficient plug-and-play component. Extensive evaluations across challenging benchmarks such as V*Bench and HRBench demonstrate that our approach consistently achieves state-of-the-art accuracy, surpassing both conventional training-free cropping strategies and reinforcement learning-based alternatives. By replacing memory-intensive multi-pass searches with a streamlined one-shot attention extraction and targeted tensor offloading pipeline, HiDe reduces peak GPU memory consumption by approximately 75%, which substantially lowers the hardware barrier for high-resolution multimodal inference.

## Impact Statement

This work does not involve any ethical concerns. All datasets used are sourced from publicly available and open-access repositories, and the models employed are also derived from open-source projects. The research is conducted purely for academic purposes, with no commercial applications or interests involved. We adhere to standard scientific practices in data usage and model evaluation, ensuring transparency, fairness, and respect for intellectual property. This paper presents work whose goal is to advance the field of machine learning. There are many potential societal consequences of our work, none of which we feel must be specifically highlighted here.

**Acknowledgement.** This work was supported by alibaba Group through Alibaba Research Intern Program.

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

## A. Ablation on the hyper-parameters

To evaluate the sensitivity and robustness of HiDe, we conduct extensive grid-search ablation experiments on the $V^*$ Bench dataset using two primary base models: Qwen2.5-VL 7B and InternVL3 8B. We focus on two core hyper-parameters: the Gaussian smoothing kernel width $\sigma$ (ranging from 1 to 3) and the binarization threshold $\alpha$ (ranging from 0.1 to 0.9).

As shown in Table 6 and Table 7, HiDe exhibits a stable performance plateau across a wide range of settings. For Qwen2.5-VL, the optimal performance is achieved at $\sigma = 1/3$ and $\alpha = 0.5/0.6$, while InternVL3 performs best at $\sigma = 1/2$ and $\alpha = 0.4/0.6$. Notably, even with suboptimal parameters, HiDe consistently outperforms the base models. The results demonstrate that HiDe is not overly sensitive to hyper-parameter tuning. The consistency across different $\sigma$ and $\alpha$ values proves that our Token-wise Attention Decoupling (TAD) effectively captures the intrinsic alignment between question semantics and visual regions, ensuring the reliability of the method in diverse scenarios.

*Table 6.* Qwen2.5-VL 7B hyper-parameters ablation experiments.

| $\sigma$ | $\alpha$ | Attr | Spatial | Avg | $\sigma$ | $\alpha$ | Attr | Spatial | Avg | $\sigma$ | $\alpha$ | Attr | Spatial | Avg |
|---|---|---|---|---|---|---|---|---|---|---|---|---|---|---|
| 1 | 0.1 | 91.3 | 81.6 | 87.4 | 2 | 0.1 | 87.8 | 80.3 | 84.8 | 3 | 0.1 | 87.8 | 76.3 | 83.2 |
| 1 | 0.2 | 93.0 | 78.9 | 87.4 | 2 | 0.2 | 91.3 | 84.2 | 88.5 | 3 | 0.2 | 90.4 | 77.6 | 85.3 |
| 1 | 0.3 | 93.9 | 86.8 | 91.1 | 2 | 0.3 | 92.2 | 84.2 | 89.0 | 3 | 0.3 | 88.7 | 84.2 | 86.9 |
| 1 | 0.4 | 93.0 | 88.2 | 91.1 | 2 | 0.4 | 93.9 | 86.8 | 91.1 | 3 | 0.4 | 91.3 | 86.8 | 89.5 |
| 1 | 0.5 | 94.8 | 88.2 | 92.1 | 2 | 0.5 | 93.0 | 84.2 | 89.5 | 3 | 0.5 | 93.0 | 84.2 | 89.5 |
| 1 | 0.6 | 93.9 | 89.5 | 92.1 | 2 | 0.6 | 94.8 | 88.2 | 92.1 | 3 | 0.6 | 92.2 | 86.8 | 90.1 |
| 1 | 0.7 | 91.3 | 88.2 | 90.1 | 2 | 0.7 | 93.0 | 85.5 | 90.1 | 3 | 0.7 | 94.8 | 88.2 | 92.1 |
| 1 | 0.8 | 93.0 | 88.2 | 91.1 | 2 | 0.8 | 93.9 | 89.5 | 92.1 | 3 | 0.8 | 94.8 | 85.5 | 91.1 |
| 1 | 0.9 | 87.8 | 84.2 | 86.4 | 2 | 0.9 | 92.2 | 88.2 | 90.6 | 3 | 0.9 | 92.2 | 92.1 | 92.1 |

*Table 7.* InternVL3 8B hyper-parameters ablation experiments.

| $\sigma$ | $\alpha$ | Attr | Spatial | Avg | $\sigma$ | $\alpha$ | Attr | Spatial | Avg | $\sigma$ | $\alpha$ | Attr | Spatial | Avg |
|---|---|---|---|---|---|---|---|---|---|---|---|---|---|---|
| 1 | 0.1 | 87.8 | 82.9 | 85.9 | 2 | 0.1 | 86.1 | 82.9 | 84.8 | 3 | 0.1 | 85.2 | 77.6 | 82.2 |
| 1 | 0.2 | 92.2 | 86.8 | 90.1 | 2 | 0.2 | 90.4 | 84.2 | 88.0 | 3 | 0.2 | 88.7 | 88.2 | 88.5 |
| 1 | 0.3 | 88.7 | 86.8 | 88.0 | 2 | 0.3 | 89.6 | 85.5 | 88.0 | 3 | 0.3 | 90.4 | 86.8 | 89.0 |
| 1 | 0.4 | 92.2 | 90.8 | 91.6 | 2 | 0.4 | 90.4 | 86.8 | 89.0 | 3 | 0.4 | 92.2 | 89.5 | 91.1 |
| 1 | 0.5 | 89.6 | 92.1 | 90.6 | 2 | 0.5 | 91.3 | 88.2 | 90.1 | 3 | 0.5 | 92.2 | 86.8 | 90.1 |
| 1 | 0.6 | 92.2 | 86.8 | 90.1 | 2 | 0.6 | 92.2 | 90.8 | 91.6 | 3 | 0.6 | 89.6 | 86.8 | 88.5 |
| 1 | 0.7 | 90.4 | 88.2 | 89.5 | 2 | 0.7 | 93.0 | 85.5 | 90.1 | 3 | 0.7 | 91.3 | 89.5 | 90.6 |
| 1 | 0.8 | 88.7 | 86.8 | 88.0 | 2 | 0.8 | 90.4 | 89.5 | 90.1 | 3 | 0.8 | 90.4 | 90.8 | 90.6 |
| 1 | 0.9 | 88.7 | 85.5 | 87.4 | 2 | 0.9 | 89.6 | 89.5 | 89.5 | 3 | 0.9 | 87.8 | 85.5 | 86.9 |

## B. Visualization of attention from semantic units at different layers to the image

To investigate how the model's internal perception of the question evolves through its architecture, we visualize the attention maps generated by specific semantic units (e.g., "dress", "apple logo") across different layers of Qwen2.5-VL 7B as shown in Fig. 8 and Fig. 9.

In initial Layers, attention is typically chaotic or uniformly distributed, indicating that the model has not yet established a strong cross-modal alignment for specific tokens. In middle Layers: the attention maps become highly explicit and localized, pinpointing the exact target regions mentioned in the query. This stage represents the peak of visual-semantic grounding. In final Layers, we observe the emergence of new artifacts or a shift toward "visual sinks" (often the corners of the image), as the model transitions from perception to final token generation.

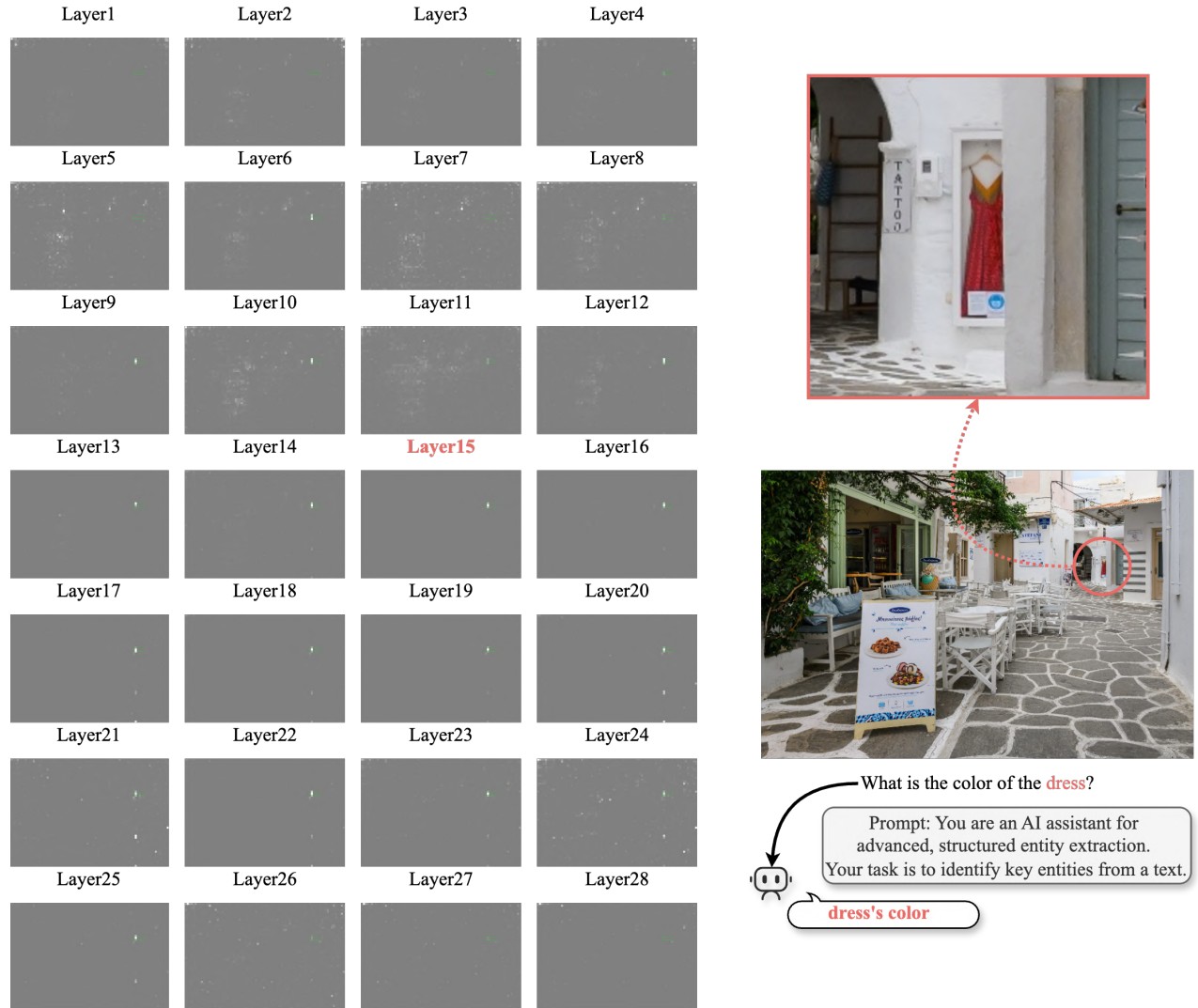

*Figure 8.* A sample from V*Bench focusing on the color of the dress.

This visualization justifies our strategy of selecting middle layers (e.g., Layer 15 or 17) for TAD. It proves that the "implicit knowledge" needed for precise localization already exists within the pre-trained model and can be successfully extracted without further training.

## C. Token-to-image attention maps analysis on InternVL3-8B

We compare our Token-wise Attention Decoupling (TAD) with the traditional "first-token attention" approach (represented by ViCrop) using the InternVL3-8B model as shown in Fig. 10.

We found that using only the first generated token often results in a "hit-or-miss" scenario, where the model may only focus on one object or incorporate significant background noise, failing in multi-target scenes. By decomposing the question into individual semantic units, TAD generates a "coverage map" where each relevant object is independently and accurately localized. For instance, in complex queries involving multiple attributes, each noun/adjective pair successfully activates its corresponding visual region.

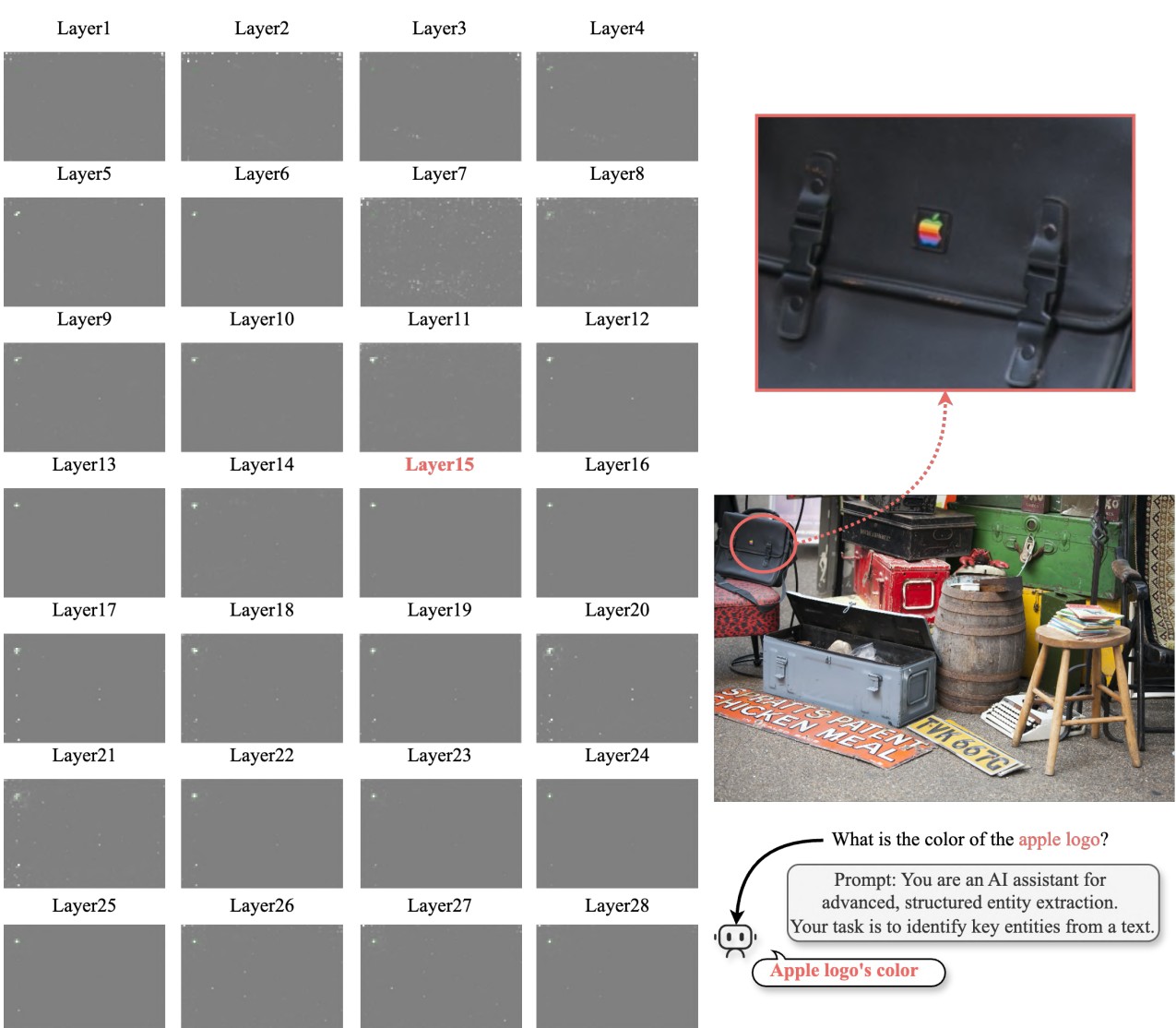

*Figure 9.* A sample from V*Bench focusing on the color of the apple logo.

Question: Is the dog on the left or right side of the scooter?

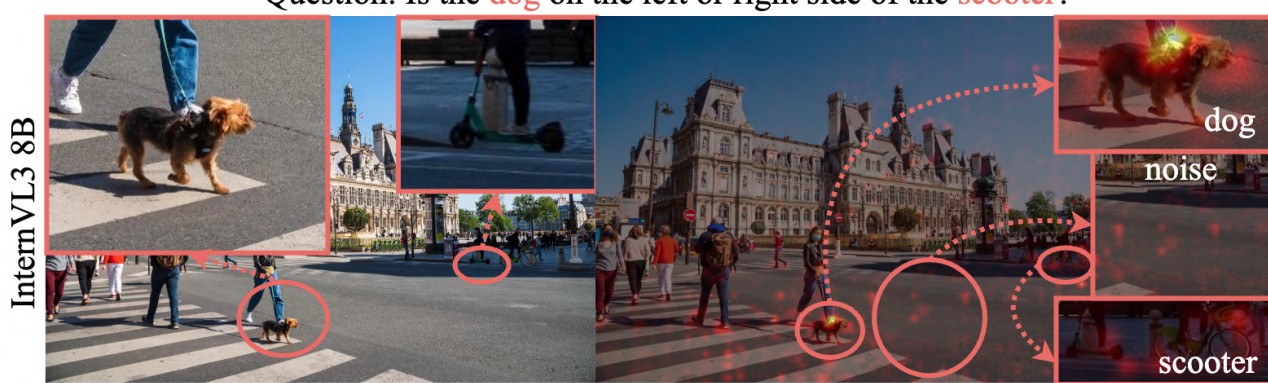

(a) First Answer Token Attention

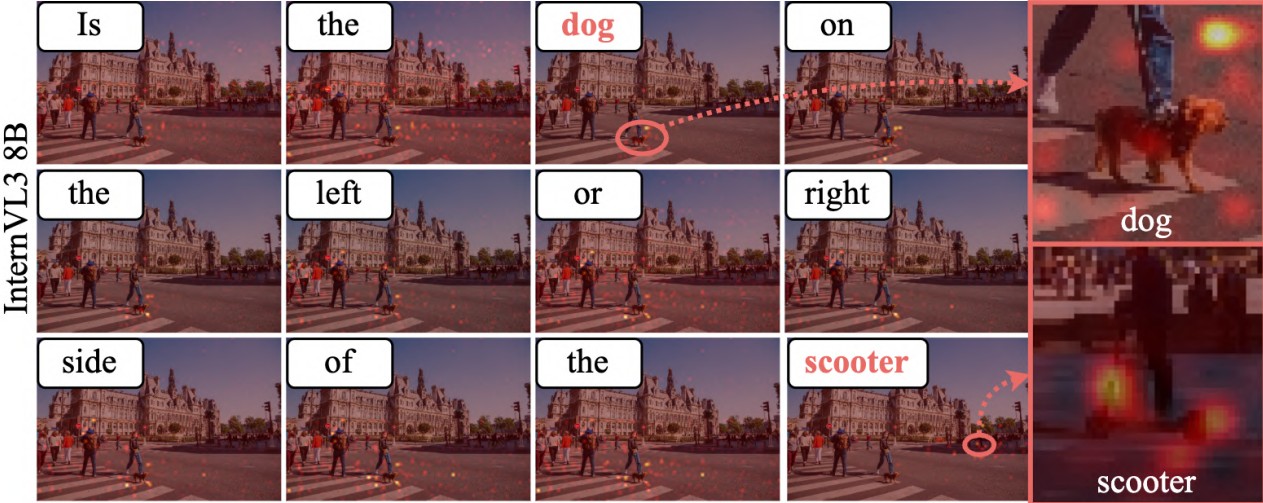

(b) Decouple Token Attention

*Figure 10.* Token-to-image attention maps. (a): attention from the first generated answer token; (b): attention maps for individual input question tokens. Using InternVL3 8B.

## D. More results on other Benchmarks

To further validate the generalization and robustness of the proposed framework, we extend our evaluation to a wider range of benchmarks that include POPE, MME-RealWorld, DocVQA, and COD10K. These experiments focus on diverse challenges such as object hallucination and document reasoning, which allows us to observe how HiDe performs when the task requirements shift from high-resolution search to general perception.

POPE (9,000 samples) (Li et al., 2023b) consists of three subsets: adversarial (Adv) (3,000 samples), popular (Pop) (3,000 samples), and random (Ran) (3,000 samples). MME-RealWorld-Lite (MME-RW-Lite) (1,919 samples) (Zhang et al., 2025b) contains two subsets: Perception (Per) (1,169 samples) and Reasoning (Rea) (750 samples) and MME-RealWorld-English (MME-RW-EN) (23,609 samples) (Zhang et al., 2025b) contains two subsets: Perception (Per) (20,767 samples) and Reasoning (Rea) (2,842 samples) (Zhang et al., 2025b). We report the accuracy on each of these subsets. Otherwise, we conducted experiments on DocVQA (Mathew et al., 2020) using ViCrop's evaluation files. We also constructed POPE-style questions using the validation set of the camouflaged object detection dataset COD10K (Fan et al., 2020) to evaluate performance in complex scenes. Specifically, for each image, we generated two questions: one asking about objects present in the image and another asking about objects absent from the image. This resulted in a total of 4,022 questions across 2,011 images.

As shown in Table 8, our method achieves improvements across different benchmark. These findings collectively confirm that our method is not only suitable for high-resolution benchmarks but also serves as a versatile enhancement for general

*Table 8.* We report answer accuracy for multiple MLLMs on other VQA tasks. The best score is highlighted in **bold**, and the second is underlined.

| Method | Model | POPE | | | | MME-RW-Lite | | | MME-RW-EN | | | DocVQA | COD10K |
|---|---|---|---|---|---|---|---|---|---|---|---|---|---|
| | | Adv | Pop | Ran | Avg | Per | Rea | Avg | Per | Rea | Avg | Acc | Acc |
| - | GPT-4o | - | - | - | - | 49.1 | **42.1** | 46.4 | 46.4 | 42.3 | 45.2 | - | - |
| - | Qwen2.5-VL 7B | 84.0 | 84.4 | 85.1 | 84.5 | 51.6 | 39.3 | 46.8 | 64.3 | 40.1 | 61.4 | 81.7 | 91.9 |
| Vicrop | Qwen2.5-VL 7B | 84.9 | 85.1 | 85.7 | 85.2 | 55.6 | 41.6 | 50.1 | 65.1 | 42.0 | 62.3 | **81.8** | 90.5 |
| **HiDe** | Qwen2.5-VL 7B | **85.1** | **85.4** | **86.3** | **85.6** | **57.8** | **42.1** | **51.7** | **66.7** | **42.9** | **63.8** | **81.8** | **92.1** |
| Δ (*vs* Qwen2.5-VL 7B) | | +1.1 | +1.0 | +1.2 | +1.1 | +6.2 | +2.8 | +4.9 | +2.4 | +2.8 | +2.4 | +0.1 | +0.2 |

*Table 9.* HiDe's final performance varies depending on the selected layer.

| Model | Method | Layer | V*(Attr) | V*(Spatial) | V*(Avg) |
|---|---|---|---|---|---|
| Internvl3 8B | - | - | 81.7 | 78.9 | 80.6 |
| Internvl3 8B | ViCrop | default-22 | 88.7 | 75.0 | 83.3 |
| Internvl3 8B | HiDe | 15 | 89.6 | 86.8 | 88.5 |
| Internvl3 8B | HiDe | 16 | 91.3 | 85.5 | 89.0 |
| Internvl3 8B | HiDe | default-17 | **92.2** | **90.8** | **91.6** |
| Internvl3 8B | HiDe | 18 | 87.0 | 85.5 | 86.4 |
| Qwen2.5-VL 7B | - | - | 80.9 | 76.3 | 79.1 |
| Qwen2.5-VL 7B | ViCrop | default-22 | 89.6 | 71.1 | 82.2 |
| Qwen2.5-VL 7B | HiDe | 14 | 87.0 | 82.9 | 85.3 |
| Qwen2.5-VL 7B | HiDe | default-15 | **94.8** | **88.2** | **92.1** |
| Qwen2.5-VL 7B | HiDe | 16 | 93.0 | **88.2** | 91.1 |
| Qwen2.5-VL 7B | HiDe | 17 | 89.6 | 85.5 | 88.0 |

*Table 10.* The layer ultimately selected by HiDe varies depending on the number of samples chosen.

| Sample Numbers | 1 | 5 | 10 | 20 | 40 | **191** |
|---|---|---|---|---|---|---|
| Qwen2.5-VL 7B | 15 | 15 | 15 | 15 | 15 | **15** |
| Internvl3 8B | 17 | 15 | 17 | 17 | 17 | **17** |

multimodal tasks because it consistently clarifies the visual input for the model.

# E. Layer Selection

We selected different layers for the two base models due to their significantly different architectures, a common practice in attention-based methods (ViCrop selects a specific 22nd layer) as shown in Table 9. Although we chose the best-performing layer, we found that performance remains robust across a range of layers. As shown in the figure, performance is optimal at the best layer and slightly degrades at suboptimal layers; however, it still achieves a significant improvement over both the original model and SOTA methods.

Additionally, we conducted a new experiment demonstrating that selecting an appropriate layer requires only a small number of samples. As shown in Table 10, just a few examples are sufficient to identify a suitable layer and achieve consistent performance.

*Table 11.* Comparative experiments with a cap on the maximum number of tokens.

| Method | Model | V* | | | HRBench4K | | | DocVQA |
|---|---|---|---|---|---|---|---|---|
| | | Attr | Spatial | Avg | FSP | FCP | Avg | Acc |
| - | Qwen2.5-VL 7B | 56.5 | 64.5 | 59.7 | 59.5 | 56.3 | 57.9 | 75.8 |
| ViCrop | Qwen2.5-VL 7B | 70.4 | 64.5 | 68.1 | 66.0 | 57.0 | 61.5 | 78.2 |
| HiDe | Qwen2.5-VL 7B | **73.0** | **73.7** | **73.3** | **71.8** | **60.0** | **65.9** | **79.6** |
| Δ (*vs* Qwen2.5-VL 7B) | | +20.5 | +9.2 | +13.6 | +12.3 | +3.7 | +8.0 | +3.8 |

*Table 12.* Results using only compact inputs. W means using both ori-image and compact image, while W/O means only using compact image.

| Model | Dataset | Base | ViCrop W | ViCrop W/O | HiDe W | HiDe W/O |
|---|---|---|---|---|---|---|
| Internvl3 8B | V* | 80.6 | 83.3 | 79.6 | **91.6** | 90.6 |
| Qwen2.5-VL 7B | V* | 79.1 | 82.2 | 74.9 | **92.1** | 90.1 |

## F. Low-resolution Experiments

Since our method is developed and analyzed under high-resolution scenarios, it is necessary to conduct experiments in low-resolution settings to examine whether HiDe introduces any overhead or negative bias in these simpler scenarios. To simulate low-resolution conditions, we explicitly cap the maximum number of visual tokens in the model input at $512 \times 28 \times 28$, thereby effectively limiting spatial resolution during inference. As shown in Table 11, under this setting, HiDe still maintains consistent improvements over baseline models, demonstrating its robustness even when high-resolution details are unavailable or unnecessary.

## G. Experiment of HiDe only compact inputs

We further evaluated HiDe on V*Bench using only compact inputs. As shown in Table 12, using only the cropped image yields slightly lower performance than combining it with the original image, confirming that global context remains important (Zheng et al., 2026; Zhang et al., 2025a). Nevertheless, it still achieves a substantial improvement over the base model.

## H. Some inference cases

We present several inference examples, including cases requiring the detection of single and multiple target regions, as shown in Fig. 11.

## I. Failure cases

As shown in the Fig. 12, HiDe incorrectly localizes the target region, leading to an incorrect answer.

## J. Algorithm of LPD

Appendix L: The Logic of Layout-Preserving Decoupling The Layout-Preserving Decoupling mechanism follows a systematic process which is designed to transform abstract attention maps into a structured visual representation. As detailed in the pseudocode shown in Fig. 13, the process begins with confidence filtering and box merging to ensure that only the most informative regions are selected for reconstruction. The core innovation of LPD is the grid-based mapping that eliminates empty background space while strictly maintaining the relative spatial positions of all identified objects.

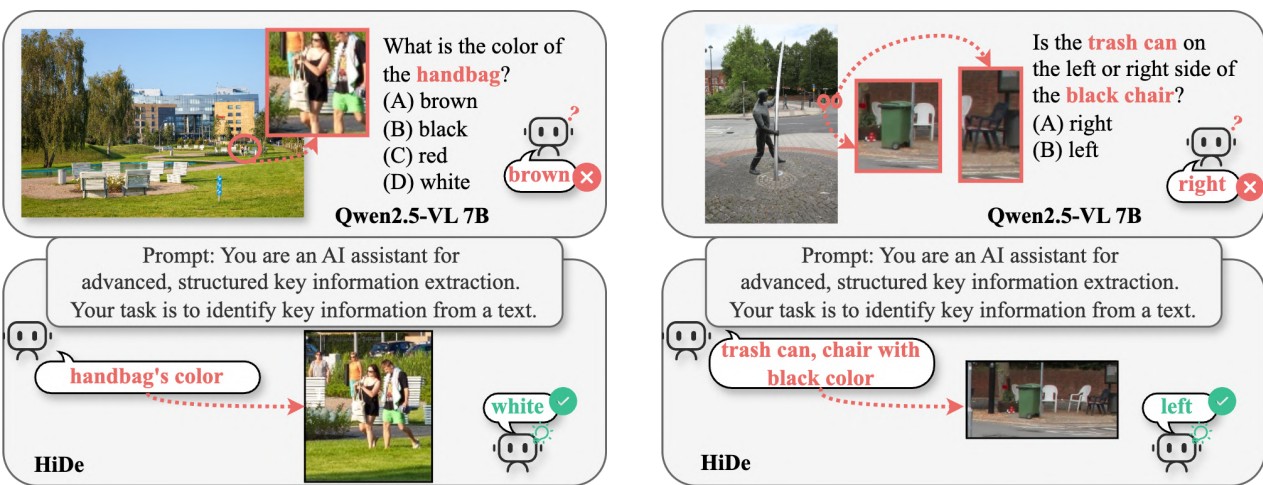

Figure 11. Left: single target region case. Right: multiple target regions case.

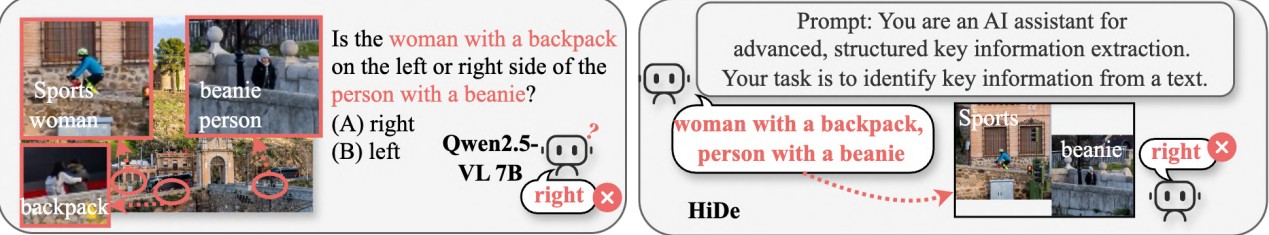

Figure 12. A fail case in V*Bench, requires locating the targets in the image and then determining the spatial relationship between them.

**Python Style pseudo code**

```
Input: I (Original Image), B_norm (Norm Boxes [x0, y0, x1, y1], (x0,y0,x1,y1∈[0,1]))

# --- STEP 1: PREPROCESSING & FILTERING ---
B_pixel = [Denormalize(b, W, H) for b in B_norm]
if not B_pixel: return None
# 1.1 Calculate Confidence via Overlap Map
OverlapMap = Array(shape=(H, W), initial_value=0)
for (x0, y0, x1, y1) in B_pixel:
    OverlapMap[y0:y1, x0:x1] += 1

# 1.2 Identify regions with at least 'n' overlaps
ThresholdMask = (OverlapMap >= n)

# 1.3 Select bboxes that intersect with the high-confidence mask
ContributingBboxes = [b for b in B_pixel if Intersects(b, ThresholdMask)]
if not ContributingBboxes: return None

# 1.4 Merge overlapping boxes and decompose by content (Alpha/Transparency)
MergedBboxes = MergeOverlapping(ContributingBboxes)
FinalBboxes = []
for b in MergedBboxes:
    FinalBboxes.extend(DecomposeByAlpha(I, b))

# --- STEP 2: LAYOUT-PRESERVING COMPACTING (Core LPD) ---
# 2.1 Extract unique grid coordinates
X_coords = Sorted(Unique(FinalBboxes.edges_x))
Y_coords = Sorted(Unique(FinalBboxes.edges_y))

# 2.2 Map X-intervals to a compact space
x_map = {X_coords[0]: 0}
new_W = 0
for i in range(len(X_coords) - 1):
    start, end = X_coords[i], X_coords[i+1]
    # Check if this column interval contains any part of a bbox
    if IsContentPresent(start, end, axis='x', bboxes=FinalBboxes):
        new_W += (end - start)
    x_map[end] = new_W

# 2.3 Map Y-intervals to a compact space
y_map = {Y_coords[0]: 0}
new_H = 0
for j in range(len(Y_coords) - 1):
    start, end = Y_coords[j], Y_coords[j+1]
    # Check if this row interval contains any part of a bbox
    if IsContentPresent(start, end, axis='y', bboxes=FinalBboxes):
        new_H += (end - start)
    y_map[end] = new_H

# --- STEP 3: RECONSTRUCTION & CENTERING ---
# 3.1 Create a compact canvas and paste patches
I_compact = NewImage(width=new_W, height=new_H, transparent=True)
for (x0, y0, x1, y1) in FinalBboxes:
    patch = Crop(I, (x0, y0, x1, y1))
    paste_pos = (x_map[x0], y_map[y0])
    Paste(I_compact, patch, at=paste_pos)

# 3.3 Return result
return I_compact
```

*Figure 13.* LPD with Confidence Filtering and Centering.

