# OpenReview forum: "HiDe: Rethinking The Zoom-IN method in High Resolution MLLMs via Hierarchical Decoupling"
_ICML.cc/2026/Conference — ICML 2026 regular_

### Official Review · Reviewer_SRdp · 2026-03-12

**Soundness:** 2
**Presentation:** 3
**Significance:** 2
**Originality:** 2
**Overall Recommendation:** 3
**Confidence:** 4

**Summary:**

This paper investigates why multimodal large language models perform poorly on high-resolution visual reasoning tasks and argues that the main issue is background interference rather than object size. Based on this analysis, the authors propose HIDE, a training-free framework that uses Token-wise Attention Decoupling to localize relevant regions using query-token attention and Layout-Preserving Decoupling to reconstruct a compact image while preserving spatial relationships. Experiments on HR-VQA benchmarks show improvements over several existing zoom-in or region-search approaches.

**Compliance With Llm Reviewing Policy:**

Affirmed.

**Final Justification:**

Due to the reasons stated in my rebuttal acknowledgement, I'm keeping my weak reject score.

**Key Questions For Authors:**

- What are the key conceptual differences between HiDE and existing attention-based zoom-in methods such as ViCrop or similar region-search approaches?
- How robust is the proposed attention-based localization mechanism across different models and prompts?
- Do the authors expect HiDe to improve performance on other multimodal tasks beyond HR-VQA, such as document or chart understanding?

**Limitations:**

Yes

**Strengths And Weaknesses:**

Strengths:
- The paper provides a useful decomposition of the commonly used zoom-in strategy into cropping and upscaling, showing that background interference plays a larger role than object size in high-resolution reasoning failures of MLLMs.
- The proposed framework can be applied at inference time without modifying or retraining the base model, making it easy to integrate with existing MLLMs.
- The method demonstrates improvement across multiple high-resolution benchmarks (e.g. V*Bench, HRBench4K, HRBench8K) and across multple base models; e.g, +13 point gain on V*Bench (Table 1, Qwen2.5-VL 7B).
- The paper proposes an optimization that significantly reduces memory overhead for attention extraction, improving practicality.

Weaknesses:
- The paper proposes a training-free framework that uses attention signals from query tokens to identify relevant regions and then reconstructs/crops those regions before feeding them back to the model. It resembles several prior approaches that aim to improve high-resolution reasoning in multimodal models through region localization. For example, previous work such as ViCrop (Zhang et al., 2025 https://arxiv.org/abs/2502.17422) and ZoomEye (Shen et al., 2024 https://arxiv.org/abs/2411.16044) also use attention-based or search-based mechanisms to identify regions of interest and perform localized reasoning. Similarly, DyFo (Li et al., 2025 https://arxiv.org/abs/2504.14920) and Divide-Conquer-Combine (Wang et al., 2025 https://arxiv.org/abs/2408.15556) propose training-free pipelines that iteratively identify relevant visual regions and feed them back into the model for improved reasoning. Compared to these works, the proposed TAD and LPD modules appear to implement a more structured version of attention-based region extraction, but the core paradigm remains similar: identify salient regions and reduce background interference before inference.
- Another key component is the use of attention maps from semantic tokens to localize relevant image regions. However, there is a substantial body of literature suggesting that attention does not always reliably correspond to visual grounding or causal explanations. For example, prior studies such as “Attention is not Explanation” (Jain & Wallace, 2019 https://arxiv.org/abs/1902.10186) and “Attention is not not Explanation” (Wiegreffe & Pinter, 2019 https://arxiv.org/abs/1908.04626) highlight the limitations of interpreting attention as a reliable indicator of importance. In the vision-language domain, several works have also shown that attention maps in multimodal transformers may capture correlations rather than true grounding signals. More recent work analyzing multimodal models, such as SelfElicit (Liu et al., 2025 https://arxiv.org/abs/2502.08767) and visual attention sink analyses (Kang et al., 2025 https://arxiv.org/abs/2503.03321), further suggests that attention distributions may exhibit systematic biases or “sink” behaviors. Given this, it would be helpful to provide additional evidence that the proposed token-wise attention decouplng reliably identifies relevant visual regions across different models and tasks.
- Experiments focus primarily on high-resolution VQA benchmarks (V*Bench and HRBench). While appropriate for the problem setting, it remains unclear whether the approach generalizes to other multimodal tasks that involve high-resolution images, such as document understanding, chart reasoning, or diagram interpretation.

Questions:
- The method extracts attention from specific intermediate layers (e.g. 15th layer for Qwen). How sensitive is the localization quality to the choice of layer? Would the method still work if different layers are used across models?
- Since the method relies on identifying semantic tokens from the input question, the extracted attention maps may depend heavily on the linguistic formulation of the query. The paper does not analyze how robust the approach is to paraphrasing or more complex queries involving multiple entities.
- In scenes with multiple relevant objects, attention from different tokens may overlap or conflict. How does the method handle such cases, and how often does this cause incorrect region proposals?
- The pipeline relies on internal attention signals of the base model. How consistent are these signals across different MLLM architectures (e.g., models with different vision encoders)?
- Some visual reasoning tasks rely on broader contextual information. Have the authors observed cases where removing background regions harms reasoning performance?

---

> ### Author Rebuttal · Authors · 2026-03-31
>
> >W1: The core paradigm remains similar to prior approaches.
>
> HiDe differs from prior region-localization methods in two aspects.
>
> First, on **signal source and memory efficiency**: ViCrop relies on attention/gradient backpropagation from the first answer token, missing objects in multi-target scenarios. Methods like ZoomEye/DyFo need multiple MLLM passes for filtering, reducing efficiency. In contrast, HiDe's TAD decouples all semantic tokens, achieving high precision with single-forward inference, surpassing ViCrop even in single-target scenarios.
>
> Second, on **spatial logic**: existing methods like ViCrop/DyFo disrupt relative positions after cropping, whereas HiDe's LPD (Algorithm 13) preserves spatial layout topologically via grid reconstruction while compressing pixels.
>
> Moreover, HiDe not only proposes a simpler, more effective method but also reveals insights about current MLLMs facing high-resolution VQA: specifically, "challenging the mainstream zoom-in assumption", which constitutes one of our key contributions.
>
> >W2: Attention is not always reliable.
>
> We agree on systematic bias and visual convergence in raw attention. Direct use fails due to attention sinks causing abnormal responses, motivating Equation 4 for purification.
>
> In TAD, we introduce a generic 'Search Prompt' to extract attention maps from non-semantic instruction tokens since they lack visual referents so their maps depict the 'pure background noise prior and convergence topology'. By subtracting this noise prior from semantic tokens' raw attention, HiDe mathematically flattens visual convergence points to extract reliable grounding signals (**see m7ez Q2 and m3WM W1 for sensitivity**).
>
> Additionally, **Appendix Table 12** provides quantitative evidence where HiDe's bounding box IoU and recall (multi-box recall up to 91.4%) prove purified attention is a highly reliable locator.
> Table 12 (Condensed)
>
> |Qwen2.5-VL 7B|HiDe|ViCrop|InternVL3 8B|HiDe|ViCrop|
> |---|---|---|---|---|---|
> |Recall|0.914|0.657|Recall|0.812|0.596|
>
>
> >W3: Whether the approach generalizes to other multimodal tasks.
>
> **Appendix E** evaluates HiDe on non-natural image tasks. **Table 8** shows HiDe improves DocVQA on Qwen2.5-VL 7B and MME-RW-Lite accuracy from 46.4% to 51.7%, enhancing capabilities in dense-text and complex scenes.
> Table 8 (Condensed)
>
> | Method | Model         | POPE | MME-RW-Lite | MME-RW-EN | DocVQA |
> | ------ | ------------- | ---- | ----------- | --------- | ------ |
> | -      | Qwen2.5-VL 7B | 84.5 | 46.8        | 61.4      | 81.7   |
> | ViCrop | Qwen2.5-VL 7B | 85.2 | 50.1        | 62.3      | 81.8   |
> | HiDe   | Qwen2.5-VL 7B | 85.6(+1.1) | 51.7(+4.9)       | 63.8(+2.4)      | 81.8(+0.1)   |
>
>
> >Q1: How sensitive is the localization quality to the choice of layer.
>
> **Table 9 (Appendix. F)** shows suboptimal layers still outperform baselines despite slight drops. **Table 10** shows that a few samples is sufficient to identify the optimal layer.
> Table 9 (Condensed)
>
> | Layer     | 14   | 15   | 16   | 17   | Base |
> | --------- | ---- | ---- | ---- | ---- | ---- |
> | V* Result | 85.3 | 92.1 | 91.1 | 88.0 | 82.2 |
>
> Table 10 (Condensed)
>
> | Sample Number | 1    | 5    | 10   | ...     | ALL  |
> | ------------- | ---- | ---- | ---- | ---- | ---- |
> | Qwen2.5-VL 7B | 15   | 15   | 15   | ...    | 15   |
> | InternVL 3 8B | 17   | 15   | 17   | ...    | 17   |
>
>
> >Q2:  How robust the approach is to paraphrasing or more complex queries involving multiple entities.
>
> Regarding this concern, **please see our response to m3WM W1** and the accompanying table. HiDe's extracted semantic tokens remain highly consistent across prompt variations, demonstrating robustness to linguistic changes.
>
> >Q3: How does the method handle scenes with multiple relevant objects.
>
> After extracting independent bounding boxes, the algorithm merges severely overlapping boxes into a unified topological component before extracting unique grid coordinates along X/Y axes, preserving topological spatial relationships even in cluttered or occluded scenarios.
>
> >Q4: How consistent are these signals across different MLLM architectures.
>
> We validated signal consistency across three distinct architectures (**Table 1**), where Qwen2.5-VL and Qwen3-VL use native-resolution ViT while InternVL3 employs patch-based ViT, all achieving SOTA results with HiDe (92.1% and 91.6%), proving such attention signals are universally present across Transformer architectures.
>
> >Q5: Do removing background regions harms reasoning performance?
>
> There seems to be a slight misunderstanding regarding our overall pipeline. **Please see our response to vZE6 W1**. Results confirm that incorporating the original image during inference yields better performance.
>
> >Key Q1: Key conceptual differences between HiDE and similar approaches?
>
> **Please refer to W1.**
>
> >Key Q2: Robustness of attention-based localization?
>
> **Please refer to Q4.**
>
> >Key Q3: Performance of HiDe on other multimodal tasks?
>
> **Please refer to W3.**

---

> > ### Author Rebuttal · Reviewer_SRdp · 2026-04-05
> >
> > I appreciate the clarifications on attention localization, sensitivity, and the extended evaluation beyond HR-VQA.
> > However, my main concern regarding novelty + contribution strength remains. The differences highlighted in the rebuttal (semantic-token attention, single-pass efficiency, layout-preserving reconstruction) appear to be incremental refinements within an existing zoom-in/region-localization paradigm, rather than a new approach. The overall pipeline, which uses query-conditined attention to identify regions and reprocess them, remains conceptually similar to prior methods. In addition, while the extended experiments are helpful, the improvements beyond HR-VQA appear modest and somewhat inconsistent, suggesting limited generality.
> > Overall, the rebuttal strengthens the empirical support but does not substantially change my assessment of the paper's contribution. I keep my weak reject score.

---

> > > ### Author Response · Authors · 2026-04-05
> > >
> > > Thank you for your thoughtful feedback and for recognizing the empirical strengths of our work.
> > >
> > > We respectfully wish to clarify three aspects that we believe distinguish our contribution.
> > >
> > > 1.Our work fundamentally reconstructs the zoom-in paradigm through hierarchical decoupling analysis. **We establish that background interference constitutes the dominant failure mode in high-resolution MLLMs.** This diagnostic insight directly challenges conventional zoom-in approaches, which rely on heuristic cropping without addressing the underlying interference mechanism. **We believe this analytical contribution redirects the community toward the actual root cause of high-resolution reasoning failures, thereby enabling more principled research that transcends iterative crop-and-scale strategies.**
> > >
> > > 2.Guided by the conclusion that removing interference is key, we systematically identify which components of traditional zoom-in methods are essential and which introduce redundancy, as shown in Figure 1.
> > >
> > > First, HiDe introduces Layout-Preserving Decoupling (LPD), a novel reconstruction mechanism that maintains topological spatial relationships while filtering out irrelevant context. This design directly addresses the limitation of ViCrop in multi-object scenarios. **To our knowledge, this represents the first introduction of such a mechanism in the literature.**
> > >
> > > Second, HiDe incorporates Token-wise Attention Decoupling (TAD), which more efficiently suppresses background regions by isolating task-relevant attention signals. **This enables more precise region selection without relying on iterative search or multiple forward passes.**
> > >
> > > For both modules, we conduct extensive ablation studies and quantitative analyses in the appendix to rigorously verify their individual and synergistic effects in reducing interference while preserving task-relevant information.
> > >
> > > **HiDe delivers SOTA accuracy with the fastest inference speed among all compared methods**, substantially outperforming approaches that rely on costly search procedures or require dataset construction for additional training.
> > >
> > > 3.**The generalizability of HiDe is supported by consistent gains across diverse benchmarks.** As shown in **Appendix Tables 8 and 11**, HiDe yields improvements on multimodal datasets that span different domains and difficulty levels. Even on DocVQA, where background clutter is relatively limited, HiDe still delivers measurable gains. On more complex real-world datasets, the advantages become more pronounced because background suppression plays a larger role. **These results suggest that the decoupling principle underlying HiDe is broadly applicable beyond the HR-VQA setting.**
> > >
> > > **We believe the analytical insight, the methodological design, and the empirical validation reinforce one another.** In the final version, we will more explicitly foreground this three-part contribution structure in the introduction and conclusion so that the scope of novelty is communicated with greater clarity.
> > >
> > > We sincerely appreciate your time and constructive critique. **If the clarifications above help address your concerns regarding the paper's contributions, we would be grateful if you might consider adjusting your score accordingly.**

---

### Official Review · Reviewer_m7eZ · 2026-03-12

**Soundness:** 3
**Presentation:** 3
**Significance:** 3
**Originality:** 3
**Overall Recommendation:** 4
**Confidence:** 4

**Summary:**

This work investigates the underperformance of Multimodal Large Language Models (MLLMs) on high-resolution images. Contrary to prior work that attributes this issue to perceptual constraints and uses zoom-in strategies, the authors conduct a hierarchical decoupling analysis and identify background interference, including semantic distractors and token-level redundancy, as the actual bottleneck. They propose HiDe, a training-free, plug-and-play framework that employs Token-wise Attention Decoupling (TAD) to refine token-level attention for fine-grained cross-modal alignment and target localization, along with Layout-Preserving Decoupling (LPD) to extract salient regions and reconstruct compact images while preserving spatial layout and removing irrelevant background. Extensive experiments show that HiDe achieves state-of-the-art performance on V*Bench, HRBench4K, and HRBench8K, while reducing peak GPU memory usage by 75% compared to previous training-free approaches.

**Compliance With Llm Reviewing Policy:**

Affirmed.

**Final Justification:**

All my concerns have been solved properly, I thus keep my initial positive score as my final recommendation.

**Key Questions For Authors:**

1. How does the grid-based spatial reconstruction in LPD handle heavily overlapping bounding boxes and complex occlusions, and can its layout preservation be considered robust in cluttered scenes?

2. How sensitive is the attention purification process to the wording of the generic “search” prompt, and would minor prompt variations lead to significantly different attention maps?

3. Is there a principled, dynamic way to select the optimal attention layer for each query or image, rather than relying on fixed manual selection?

**Limitations:**

yes

**Strengths And Weaknesses:**

Strengths:

1. The hierarchical decoupling analysis provides novel insights, challenging the mainstream zoom-in assumption and correctly identifying background interference and token redundancy as the key bottlenecks for high-resolution MLLMs.

2. The proposed HiDe framework is well-designed, training-free, and model-agnostic. It achieves state-of-the-art performance across benchmarks while drastically reducing GPU memory usage, offering strong practical value for real-world deployment.

3. The paper is well-structured and clearly written, with convincing visualizations that effectively illustrate the working mechanism of the TAD module and support the core claims.

Weaknesses:
1. It seems that the method relies on handcrafted heuristics and fixed hyperparameters, lacks adaptive mechanisms, and faces potential robustness issues under complex spatial scenes (occlusions, dense objects) while being constrained by the base model’s inherent visual-semantic grounding ability.

2. Aggressive background pruning may risks losing global visual context. Besides, critical implementation details for semantic token extraction are underspecified, which would harm reproducibility.

3. Token-wise attention processing may introduces query-length-dependent computational overhead.

4. The baseline comparison is limited to ViCrop and DeepEyes, excluding recent dynamic-resolution and patch-selection methods such as LLaVA-UHD and Matryoshka-based approaches, weakening the justification of its state-of-the-art claim. Meanwhile, the evaluation setting for proprietary models like GPT-4o and OpenAI o3 is ambiguous, as it is unspecified whether they used the same zoom-in strategy, leading to an unfair comparison.

---

> ### Author Rebuttal · Authors · 2026-03-31
>
> >W1: The method relies on handcrafted heuristics and fixed hyperparameters.
>
> We argue that our method’s effectiveness is not dependent on the fixed hyperparameters. **Appendix A** grid searches for $\sigma$ and $\alpha$ show HiDe maintains stable performance across broad parameter ranges, where it consistently outperforms the base model even with suboptimal settings.
>
> Table 5 (Condensed)
>
> | σ=1 (α/Avg) | σ=2 (α/Avg) | σ=3 (α/Avg) |
> |-------------|-------------|-------------|
> | 0.3/91.1    | 0.4/91.1    | 0.6/90.1    |
> | 0.4/91.1    | 0.5/89.5    | 0.7/92.1    |
> | 0.5/92.1    | 0.6/92.1    | 0.8/91.1    |
> | 0.6/92.1    | 0.7/90.1    | 0.9/92.1    |
>
> Our main finding is that current large multimodal models already contain strong fine-grained grounding ability: mid-to-deep layers produce accurate localization signals, and errors often arise because these signals are weakened by background noise. HiDe therefore acts as a signal amplifier to expose this latent capability, and its background-suppression mechanism consistently brings additional gains as base models improve (see **Table m3WM-W2**).
>
> For complex spatial scenes, HiDe further introduces specialized modules. TAD decouples token-level attention to support localization under dense objects and occlusion, while LPD can recover bounding boxes without complete contours by relying on strong regional attention responses and connected-component analysis.
>
> >W2: Aggressive background pruning may risks losing global visual context. Critical implementation details for semantic token extraction are underspecified
>
> (**see vZE6-W1**) Global context is preserved because we feed both original images and $I_{compact}$ during inference. Semantic token extraction details (**see m3WM-W1**) and code are in supplementary materials, where a complete version will be released upon acceptance.
>
> >W3: Token-wise attention processing may introduces query-length-dependent computational overhead.
>
> As presented in **Table 4**, the inference time for our HiDe method (14 minutes) is approximately double that of the baseline Qwen2.5-VL model (7 minutes). This increased latency is an expected consequence of our two-pass design not our proposed attention mechanism. The attention computation itself is highly efficient, employing a single layer with few text-token queries.
>
> >W4: Missing recent methods such as LLaVA-UHD. Evaluation setting for gpt-4o is ambiguous
>
> HiDe is orthogonal to dynamic resolution methods like LLaVA-UHD, which reduce input tokens for arbitrary sizes, whereas HiDe mitigates background interference in attention to boost high-resolution VQA with modest overhead.
>
> We additionally include evaluation results for LLaVA-UHD v3 [1] here:
>
> | Method        | V* Attr/Spatial/Overall      | HR-4K FSP/FCP/Avg |
> |---------------|---------------|-------------------|
> | LLaVA-UHD-V3  | 66.1/65.8/66.0| 73.8/58.0/65.9    |
> | Qwen2.5-VL 7B | 80.9/76.3/79.1| 88.8/54.8/71.8    |
> | +ViCrop       | 89.6/71.1/82.2| 90.5/57.5/74.0    |
> | +HiDe         | 94.8/88.2/92.1| 95.5/59.5/77.5    |
>
> For GPT-4o/o3, we use default high-resolution APIs with inputs identical to Qwen2.5-VL so that the pipeline remains unmodified, although internal operations are inaccessible since GPT models are closed-source.
>
> >Q1: How does LPD handle heavily overlapping bounding boxes?
>
> After independent bounding boxes are extracted, severely overlapping ones are merged into unified topological components before unique X/Y grid coordinates are extracted, which preserves spatial relationships even in cluttered or occluded scenarios.
>
> >Q2: How sensitive is Hide to the search prompt?
>
> **Equation 4** subtracts a "background noise prior" from attention maps of non-semantic tokens such as "Search" or "the", because tokens that lack visual referents exhibit noise-like attention patterns. Consequently, background noise topology remains consistent across prompts such as "Search", "Find", or "Look for", as validated on V\* Bench:
>
> | Method | Rewrite                  | Attr | Spatial | Avg  |
> | ------ | ------------------------ | ---- | ------- | ---- |
> | base   | -                        | 80.9 | 76.3    | 79.1 |
> | HiDe   | Search the following ... | 94.8 | 88.2    | 92.1 |
> |  HiDe  | Find the following ...   | 94.8 | 88.2    | 92.1 |
> |   HiDe | Looking for following... | 92.2 | 86.8    | 90.1 |
>
> >Q3: A dynamic way to select the optimal attention layer?
>
> Our observations show that the optimal layer is mainly determined by model architecture rather than specific inputs, so a fixed layer can capture grounding signals consistently across scenarios. After statistical calibration, no further adjustment is needed, and it generalizes well to data from other sources. As shown in **Table 10 (Appendix F)**, only a small pilot run is sufficient to identify the optimal layer, avoiding extensive data collection.
>
> [1] LLaVA-UHD v3: Progressive Visual Compression for Efficient Native-Resolution Encoding in MLLMs

---

> > ### Author Rebuttal · Reviewer_m7eZ · 2026-04-04
> >
> > All my concerns have been solved properly, I thus keep my initial postive score as my final recommendation.

---

> > > ### Author Response · Authors · 2026-04-04
> > >
> > > Thank you for your constructive feedback and your positive evaluation of our work. We are glad our clarifications have addressed your concerns. We will incorporate your suggestions to further improve the manuscript.

---

### Official Review · Reviewer_m3WM · 2026-03-15

**Soundness:** 2
**Presentation:** 1
**Significance:** 2
**Originality:** 2
**Overall Recommendation:** 4
**Confidence:** 3

**Summary:**

The paper proposes the Hierarchical Decoupling Framework (HiDe), which aims at increasing performance on the high-resolution VQA tasks by manipulating the input space. More specifically, the method uses attention from the semantic tokens to localize important regions in the image (TAD) and isolates these regions while preserving the spatial information (LPD).

**Compliance With Llm Reviewing Policy:**

Affirmed.

**Final Justification:**

My concerns and questions have been mostly resolved, I've increased the score by 1

**Key Questions For Authors:**

From the example of the search prompt in Figure 6, it is unclear whether the prompt also contains semantic tokens (e.g., “motorcycle”, “dog”). Do these tokens contribute to the average attention that is subtracted in Eq. (4)?
How are semantic tokens extracted from the question?
Would it be possible for the authors to show how the method scales to larger models? Does it still have gains over the base models when the models are more capable?

**Limitations:**

yes

**Strengths And Weaknesses:**

Strengths
The method demonstrates consistent gains over the baseline models as well as a “zoom-in” method ViCrop 3 models and several datasets.
The proposed method is training-free and model-agnostic, making it potentially applicable to multiple architectures.
The authors introduce a trick to reduce the RAM usage that applies to both HiDe and ViCrop.

Weaknesses
Several aspects of the method are insufficiently specified, which makes reproducibility difficult. The examples are:
The definition and construction of the generic “search prompt” used in TAD, and the sensitivity of the method to that prompt.
The procedure used to identify semantic tokens from the input question.
The experimental evaluation appears somewhat unsystematic. For example, ViCrop results are reported for InternVL3-8B and Qwen2.5-VL-7B but not for Qwen3-VL-8B, which is the model with the smallest reported gains. Similarly, results for Qwen2.5-VL-32B are shown without corresponding HiDe or ViCrop comparisons. Clarifying these omissions would improve the consistency of the evaluation.
HiDe operates by modifying the input representation and requires additional processing steps, including two forward passes. The reported inference time is roughly double that of the base model, raising questions about practicality in real-world deployment. At the same time, improvements from stronger base models (e.g., the performance gap between Qwen2.5-VL and Qwen3-VL) suggest that scaling the base model itself can substantially improve performance. It would therefore be useful to better understand the regime in which HiDe provides advantages over simply using a more capable base model under similar compute or latency constraints.
The paper reports model-specific hyperparameter tuning (σ, α) for HiDe, but does not describe whether similar tuning was performed for the ViCrop baseline. Since attention-based localization methods can be highly sensitive to these parameters, it is unclear whether the baseline results reflect optimal configurations. Clarifying whether ViCrop was tuned under the same protocol would improve the fairness and reproducibility of the comparison. Additionally, the layer ablation shows that the method has high sensitivity to layer selection.

---

> ### Author Rebuttal · Authors · 2026-03-31
>
> We thank you for your constructive feedback on HiDe gains, training-free nature, and memory optimizations.
>
> >W1: Insufficiently specified method makes reproducibility difficult. Prompt sensitivity?
>
> We will clearify the details in the revised manuscript. We also have added code to the supplementary material for one-click experiment reproduction. The search prompt is now fixed as:
>
> ```You are a highly precise language analysis engine. Your sole function is to extract entities (e.g., objects, people)from a user'question. Return a single line wrapped in \<FINAL_OUTPUT> and <\/FINAL_OUTPUT>.```
>
> For prompt sensitivity, we tested V*Bench with Qwen3-Max using three rewrite styles (passive, synonym, chatty). HiDe's extracted semantic tokens remain highly consistent across variations, demonstrating robustness to linguistic changes.
>
> Rewrite Examples
>
> - Original: Is the woman with a backpack on the left or right side of the person with a beanie?
> - Passive: On which side of the person with a beanie is the woman with a backpack positioned, left or right?
> - Synonym: Is the female with a backpack on the leftward or rightward side of the individual with a beanie?
> - Chatty: I would be grateful if you could tell me whether the woman with a backpack is on the left or right side of the person with a beanie.
>
> | Rewrite | base(Attr/Spt/Avg) | HiDe(Attr/Spt/Avg) |
> |---------|--------------------|--------------------|
> | -       | 80.9/76.3/79.1     | 94.8/88.2/92.1     |
> | chatty  | 82.6/75.0/79.6     | 94.8/85.5/91.1     |
> | passive | 84.1/81.6/83.1     | 93.9/86.8/91.1     |
> | synonym | 80.0/75.0/78.0     | 93.9/84.2/90.1     |
>
> >W2: Unsystematic experimental evaluation.
>
> We added ViCrop results for Qwen3-VL-8B and Qwen2.5-VL-32B, where HiDe gains are orthogonal to base model capability so significant improvements persist even with stronger models.
>
> | Model | Method | Attr | Spat | V* (Avg) | FSP | FCP | HR-4K(Avg) | FSP | FCP | HR-8K(Avg) |
> |--------------|--------|------|------|----------|-----|-----|------------|-----|-----|------------|
> | Qwen2.5-VL 3B | Base | 80.9 | 61.8 | 73.3 | 83.0 | 50.3 | 66.6 | 79.8 | 45.0 | 62.4 |
> | | ViCrop | 81.7 | 65.8 | 75.4 | 86.3 | 49.8 | 68.0 | 80.3 | 45.5 | 62.9 |
> | | HiDe | 85.2 | 72.4 | 80.1(+6.8) | 87.8 | 51.5 | 69.6(+3.0) | 88.3 | 51.3 | 69.8(+7.4) |
> | Qwen2.5-VL 7B | Base | 80.9 | 76.3 | 79.1 | 88.8 | 54.8 | 71.8 | 84.2 | 51.5 | 67.9 |
> | | ViCrop | 89.6 | 71.1 | 82.2 | 90.5 | 57.5 | 74.0 | 84.5 | 56.3 | 70.4 |
> | | HiDe | 94.8 | 88.2 | 92.1(+13.0) | 95.5 | 59.5 | 77.5(+5.7) | 86.8 | 58.5 | 72.6(+4.7) |
> | Qwen2.5-VL 32B | Base | 87.8 | 88.1 | 87.9 | 89.8 | 58.0 | 73.9 | 84.5 | 56.3 | 70.4 |
> | | ViCrop | 90.4 | 85.5 | 88.5 | 90.5 | 58.8 | 74.6 | 88.8 | 54.3 | 71.5 |
> | | HiDe | 91.3 | 89.5 | 90.6(+2.7) | 92.0 | 60.0 | 76.2(+2.3) | 90.5 | 57.5 | 74.0(+3.6) |
> | Qwen3-VL 8B | Base | 87.0 | 78.9 | 83.8 | 90.2 | 63.7 | 77.0 | 84.0 | 62.3 | 73.1 |
> | | ViCrop | 89.6 | 75.0 | 83.8 | 91.8 | 64.3 | 78.0 | 88.0 | 63.3 | 75.6 |
> | | HiDe | 92.9 | 85.5 | 89.5(+5.7) | 93.5 | 69.0 | 81.2(+4.2) | 96.5 | 71.2 | 83.9(+10.8) |
>
> >W3: Two forward passes is not practical.
>
> Compared with similar training-free methods or other training methods, HiDe achieves significantly faster inference. **Table 4** Shows, on the V* dataset, total inference time is 110 min for DeepEyes, 28 min for ViCrop, and only 14 min for HiDe, while achieving SOTA accuracy. We thus believe HiDe offers superior practical deployability over competing approaches.
>
> >W4: Unfair if the parameter tuning not for Vicrop.
>
> In **Table 1**, we apply the same layer selection tuning to both HiDe and ViCrop for a fair comparison. ViCrop’s lower performance is due to its structural limitation of predicting only a single bounding box, whereas HiDe’s ability to model multiple semantic regions enables stronger multi-object reasoning and better overall performance.
>
> >W5: High sensitivity to layer selection.
>
> Although layer selection is fixed, HiDe is robust and efficient: its performance does not rely on finding a perfect layer. As shown in Table 8, even sub-optimal layers still significantly outperform all baselines, indicating that the gains come from the architecture rather than sensitive tuning. Moreover, layer selection is a one-time, low-cost process for each architecture, requiring only a few validation samples (**Table 10**) and generalizing well across datasets.
>
> >Q1: Search prompt also contains semantic tokens?
>
> Semantic tokens (e.g., "motorcycle") are extracted via the LLM's reasoning from the question + extraction prompt. They are excluded from the subtracted average attention in **Eq. (4),** which computes "background noise prior" solely from non-semantic instruction tokens ("Search", "the"). This preserves high-frequency semantic signals.
>
> >Q2: Gains over more capable base models?
>
> Refer to W2. When the base model is more capable, HiDe still maintains a big advantage.

---

> > ### Author Rebuttal · Reviewer_m3WM · 2026-04-03
> >
> > My concerns have been mostly resolved.

---

> > > ### Author Response · Authors · 2026-04-04
> > >
> > > We appreciate your constructive feedback during the review process and are pleased that our clarifications have addressed your concerns. Thank you for your time and for updating your score. We will incorporate your suggestions to further improve the manuscript.

---

### Official Review · Reviewer_vZE6 · 2026-03-18

**Soundness:** 3
**Presentation:** 3
**Significance:** 3
**Originality:** 3
**Overall Recommendation:** 4
**Confidence:** 4

**Summary:**

This paper proposes HiDe that decouples 1) proposal generation of key object regions from background semantic noise and 2) relative spatial layout preservation from irrelevant context. The paper aims to solve the understanding and reasoning bottlenecks of MLLMs that are brought by high-resolution images. The performance is promising compared to baseline methods.

**Compliance With Llm Reviewing Policy:**

Affirmed.

**Key Questions For Authors:**

See weaknesses above.

**Limitations:**

yes.

**Strengths And Weaknesses:**

Strengths:

1) The paper is well analyzed, Section. 3 is insightful, it is the crop operation that leads to better understanding on challenging high-resolution images instead of the zoom-in operation popularized in common beliefs. The experiment of adjusting mask ratio and tokens of background in Figure. 4 strongly support authors' claims.
2) The proposed HiDe is simple and highly aligns the above analysis, though being simple, i.e., identifying key object regions using cross-attention maps of semantic tokens in query text to image tokens, proposing object regions by thresholding and preserves relative spatial layout on a blank canvas that prevents background distractors, these tweaks are simple but working great as in the experiment section and efficiency comparison.

Weaknesses:

1) The major concern is on questions that require reasoning over global context, since the background noise and irrelevant objects that are not mentioned in the query question are both removed, there is no way for the model to reason over these information when needed, this point is somewhat limited.

---

> ### Author Rebuttal · Authors · 2026-03-31
>
> We are very pleased that you found our analysis insightful.
>
> > W1: There is no way for the model to reason over background information when needed, this point is somewhat limited.
>
> Thank you for your question. There seems to be a slight misunderstanding regarding our overall pipeline. We kindly invite you to refer to **Line 311, page right**:
>
> ```Finally, the original image, $I_{compact}$, and the question are fed to the model for the final answer.```
>
> Indeed, relying solely on the $I_{compact}$ for reasoning would lead to the issues you are concerned about, we input **both the original image and the $I_{compact}$ together** during inference. This approach mitigates the risk of errors arising from incorrect region selection in the $I_{compact}$. Furthermore, this strategy aligns with your observation: for complex tasks requiring global context, the model can still rely on information from the original image for reasoning.
>
> In the **Appendix Table 13**, we have included experiments comparing performance with and without the original image. Additionally, we have added a new set of experiments using Qwen2.5-VL 7B as the backbone on HRBench 4K as follows, where the xxx W means with original image and the xxx W/O means without . These experiments fully demonstrate that the model still requires the original image for reasoning. Without it, the scenario you described would occur, leading to performance degradation.
>
> | Model         | Dataset    | Base | HiDe W | HiDe W/O | ViCrop W | ViCrop W/O |
> | ------------- | ---------- | ---- | ------ | -------- | -------- | ---------- |
> | Qwen2.5-VL 7B | V*         | 79.1 | 92.1   | 90.1     | 82.2     | 74.9       |
> | Qwen2.5-VL 7B | HRBench 4K | 71.8 | 77.5   | 69.6     | 74.0     | 67.5       |

---

### Decision · Program_Chairs · 2026-04-30

**Decision:**

Accept (regular)

**Comment:**

After reading the paper and the rebuttal. The AC agrees with the positive points raised by the reviewers, including the insightful analysis that challenges the mainstream zoom-in assumption (vZE6, m7eZ, SRdp) , the consistent performance improvements across multiple high-resolution benchmarks (m3WM, m7eZ, SRdp) , the training-free and model-agnostic nature of the proposed framework (m3WM, m7eZ, SRdp) , and the significant reduction in memory overhead (m3WM, m7eZ, SRdp). Since the authors' rebuttal adequately resolved the majority of the reviewers' concerns (m3WM, m7eZ) and the method offers strong practical value for real-world deployment, the AC tends to accept this paper